# Learning to Prompt for Generalizable Instance Segmentation via Bi-Level Optimization

**Li Zhang**                                                                                    *liz042@ucsd.edu*
*Department of Electrical and Computer Engineering*
*University of California, San Diego*

**Pengtao Xie**                                                                                  *p1xie@ucsd.edu*
*Department of Electrical and Computer Engineering*
*University of California, San Diego*

Reviewed on OpenReview: *https://openreview.net/forum?id=zN1yKIIVxN*

## Abstract

The Segment Anything Model has revolutionized image segmentation with its zero-shot capabilities, yet its reliance on manual prompts hinders fully automated deployment. While integrating object detectors as prompt generators offers a pathway to automation, existing pipelines suffer from two fundamental limitations: objective mismatch, where detectors optimized for geometric localization do not correspond to the optimal prompting context required by SAM, and alignment overfitting in standard joint training, where the detector simply memorizes specific prompt adjustments for training samples rather than learning a generalizable policy. To bridge this gap, we introduce BLO-Inst, a unified framework that aligns detection and segmentation objectives by bi-level optimization. We formulate the alignment as a nested optimization problem over disjoint data splits. In the lower level, the SAM is fine-tuned to minimize segmentation loss given the current detection proposals on a subset $(D_1)$. In the upper level, the detector is updated to generate bounding boxes that explicitly minimize the validation loss of the fine-tuned SAM on a separate subset $(D_2)$. This effectively transforms the detector into a segmentation-aware prompt generator, optimizing the bounding boxes not just for localization accuracy, but for downstream mask quality. Extensive experiments demonstrate that BLO-Inst achieves superior performance, outperforming standard baselines on tasks in general and biomedical domains. The code of BLO-Inst is available at `https://github.com/importZL/BLO-Inst`.

## 1 Introduction

Instance segmentation, the task of detecting and outlining individual objects in an image, is a core requirement for applications ranging from autonomous driving to biomedical analysis Yi et al. (2019); Zhou et al. (2020). Traditionally, this field relied on specialized models trained for specific tasks, such as Mask R-CNN He et al. (2017) and SOLO Wang et al. (2021). However, these approaches often suffer from limited generalization and require training with large annotated datasets. In contrast, the emergence of foundation models has fundamentally shifted this landscape Benigmim et al. (2024); Zhou et al. (2025) by leveraging their extensive prior knowledge. The Segment Anything Model (SAM) Kirillov et al. (2023) serves as a robust foundation model trained on 11 million images to handle diverse tasks without retraining. Unlike traditional methods, SAM operates as a promptable method, generating high-quality masks from prompts like points or boxes. While this design is ideal for interactive segmentation, it presents a bottleneck for automated pipelines where human input is infeasible Li et al. (2025). Consequently, deploying SAM for autonomous instance segmentation requires replacing manual guidance with a detector capable of self-generating accurate prompts.

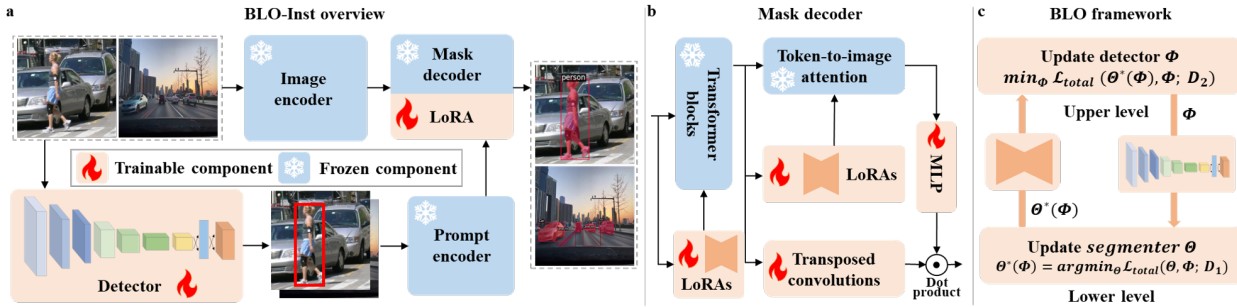

Figure 1: Overview of the BLO-Inst framework. **(a)** The architecture combines a trainable YOLO detector (parameters $\Phi$) with the Segment Anything Model (SAM). **(b)** We employ Parameter-Efficient Fine-Tuning (PEFT) by freezing SAM's heavy encoder and injecting learnable LoRA layers (parameters $\Theta$) into the mask decoder. **(c)** The Bi-Level Optimization process: the lower level updates the segmenter $\Theta$ on $D_1$ to minimize segmentation loss given fixed prompts, while the upper level updates the detector $\Phi$ on $D_2$ to generate prompts that minimize the segmenter's validation loss.

To achieve automation, a common strategy is to combine SAM with object detectors (e.g., YOLO Redmon et al. (2016) and DINO Liu et al. (2024)) in a sequential pipeline, where the detector provides bounding boxes as prompts. However, simply using a pretrained detector suffers from a fundamental objective mismatch, as the box that fits the target object perfectly is often not the best prompt for creating a good mask. For instance, as shown in Figure 2, a pedestrian might need a tighter box to remove background noise, while a cell might need a larger box to capture intact structure. To address this, recent works like USIS-SAM Lian et al. (2024) and RSPrompter Chen et al. (2024) attempt to train the detector and SAM together based on the sum of segmentation and detection losses. While this enables the detector to output desired prompts for mask decoder, it leads to another limitation: alignment overfitting. In this standard setup, the detector and segmenter are trained on the exact same data examples. This causes the detector to simply memorize the specific box adjustments needed to minimize the loss for those training samples, rather than learning a general rule for creating good prompts for the mask decoder. Consequently, when applied to new images during testing, this memorized alignment may break down, resulting in sub-optimal segmentation. Such limitation can also be experimentally observed through the performance gap (Figure 6) between "standard single-level optimization" and "bi-level optimization" in the ablation study that explore the impact on optimization strategy.

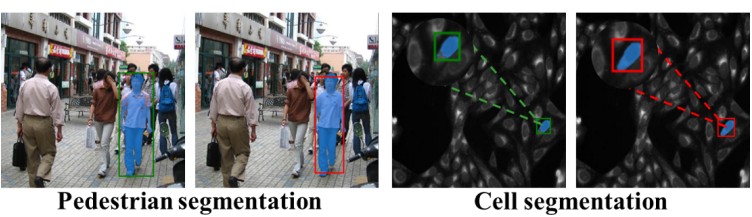

**Pedestrian segmentation**  **Cell segmentation**

Figure 2: Examples of the objective mismatch. Pedestrian (Left): A tighter box (Red) outperforms precise one (Green) by reducing background clutter. Cell (Right): A larger box (Red) outperforms precise one (Green) by providing essential context.

To address these limitations, we draw inspiration from cross-validation strategies commonly used in meta-learning and bi-level optimization Kim & Hospedales (2025). In standard joint training, if the detector is optimized on the exact same data as the segmenter, it easily overfits by memorizing the specific box adjustments needed for that training set. To prevent this, we partition the available data into two disjoint subsets. We fine-tune the segmenter to adapt to the detector's prompts on a training split, but we update the detector based on how well the adapted segmenter performs on a separate validation split. This explicit cross-validation mechanism shifts the learning objective from memorizing sample-specific coordinates to learning a robust, generalizable prompting rule, thereby resolving the objective mismatch and alignment overfitting simultaneously. This shifts the learning objective from simply finding the object to finding the optimal prompt for the segmenter, thereby resolving the objective mismatch and alignment overfitting simultaneously.

Realizing this concept, we introduce BLO-Inst, a unified framework that implements this strategy via bi-level optimization (BLO) Choe et al. (2022). Specifically, as shown in Figure 1, BLO-Inst manages two sets of parameters: the segmentation model (injecting LoRA layers Hu et al. (2021) while freezing the heavy image encoder) and the object detector. Rather than standard joint training, we formulate the process as a nested optimization problem over disjoint data subsets, which we define formally in Section 3. By segregating the learning processes of the segmenter and detector onto distinct data splits, BLO-Inst effectively mitigates the risk of alignment overfitting. This ensures the detector learns a robust, generalizable prompting policy that helps SAM segment new images correctly.

Our contributions are summarized as follows:

- We identify the alignment overfitting in current automated segmentation pipelines where standard joint training causes detectors to memorize training data rather than learning generalizable prompting strategies for a specific task.

- We propose BLO-Inst, a bi-level optimization framework that formulates the detector's weights as meta-parameters. By optimizing the detector on a separate validation split, we effectively prevent alignment overfitting, ensuring the model learns robust prompting rules that generalize to unseen data rather than memorizing training distributions.

- Extensive experiments across general and biomedical domains validate that BLO-Inst achieves superior performance, outperforming standard joint-training baselines and architectural modification approaches.

## 2 Related work

### 2.1 Instance segmentation

Instance segmentation is a core computer vision challenge that unifies object detection and semantic segmentation, requiring the model to not only localize objects of interest but also delineate their precise pixel-level boundaries Liu et al. (2018b); Yang et al. (2023). Traditionally, deep learning approaches have been dominated by two-stage frameworks, epitomized by Mask R-CNN He et al. (2017). These methods operate by first generating candidate region proposals by a Region Proposal Network (RPN) Ren et al. (2015) and subsequently performing fine-grained classification and mask generation using mechanisms like RoIAlign Gong et al. (2021). While highly accurate, the sequential nature of two-stage models often incurs high computational latency. To address this, one-stage architectures such as SOLO Wang et al. (2020) and SOLOv2 Wang et al. (2021) were developed to predict masks directly from full-image feature maps without explicit proposal generation, offering a superior trade-off between inference speed and accuracy. More recently, the field has witnessed a paradigm shift towards transformer-based architectures Vaswani et al. (2017) like Mask2Former Cheng et al. (2022), which formulate segmentation as a set prediction problem using learnable queries, paving the way for the prompt-based paradigms.

With the advent of vision foundation models, recent research has shifted toward utilizing the Segment Anything Model (SAM) Kirillov et al. (2023) for instance-level tasks. While open-vocabulary detectors like GroundingDINO Liu et al. (2024) and YOLO-World Cheng et al. (2024) can be cascaded with SAM in a sequential pipeline, this leads to a disjoint optimization problem where the detector is optimized for box regression, while the segmenter requires prompt optimal to predict segmentation maps. To address this, recent approaches have proposed automated prompting modules. USIS-SAM Lian et al. (2024) introduces a lightweight prompt generator trained from scratch for underwater image processing, while RSPrompter Chen et al. (2024) attaches query-based heads to the SAM encoder for remote sensing. However, these methods rely on standard joint training strategies on the same dataset. As discussed, this approach suffers from alignment overfitting. In contrast, our work structures this alignment primarily as a nested bi-level optimization problem to ensure robust alignment. By separating the learning processes onto distinct data splits, we prevent alignment overfitting. Within this framework, the bounding boxes generated by the detector act much like dynamic hyper-parameters that guide the segmenter's response, serving as a useful analogy for how optimizing the detector on a separate validation split maximizes generalization.

## 2.2 Foundation model adaptation

The emergence of large-scale foundation models has necessitated efficient strategies to adapt their generalized representations to downstream tasks without the cost of training from scratch Radford et al. (2018); Devlin et al. (2018). Current adaptation paradigms primarily fall into two categories: Prompt Tuning and Parameter-Efficient Fine-Tuning (PEFT). Prompt tuning methods, such as CoOp Zhou et al. (2022) and VPT Jia et al. (2022), introduce learnable tokens to the input space to guide the frozen model's behavior for specific downstream tasks. Conversely, PEFT strategies like Adapters Houlsby et al. (2019) and Low-Rank Adaptation (LoRA) Hu et al. (2021) inject lightweight trainable modules into the model's architecture, updating only a fraction of the parameters. In the context of the Segment Anything Model (SAM) Kirillov et al. (2023), these strategies have been extensively explored for domain-specific applications. Approaches such as MedSAM Wu et al. (2025), SAMed Zhang & Liu (2023), and BLO-SAM Zhang et al. (2024) effectively leverage LoRA or adapter layers to adapt SAM to medical imaging modalities, significantly improving segmentation accuracy on CT and MRI data. While methods like BLO-SAM Zhang et al. (2024) and AM-SAM Li et al. (2024) pioneered the adaptation of foundation models, they are fundamentally restricted to binary semantic segmentation. These approaches primarily optimize model weights to identify pixels belonging to a specific foreground category against a background, handling multi-class scenarios only through an ensemble of multiple binary tasks. In contrast, BLO-Inst is natively designed for the more complex multi-class instance segmentation landscape. Rather than focusing on category-level pixel classification, our framework allows a single YOLO-based prompt generator to learn how to produce unique, geometrically optimized bounding boxes for multiple distinct object instances across various categories simultaneously.

Furthermore, BLO-Inst explicitly addresses the "alignment overfitting" mechanism that limits the generalization of recent automatic prompting methods like AM-SAM Li et al. (2024), CPC-SAM Miao et al. (2024), and RSPrompter Chen et al. (2024). These models typically rely on standard joint training on a single dataset, which often causes the prompt generator to memorize specific coordinate adjustments for training samples rather than learning a robust, generalizable rule. By formulating the detector's weights directly as meta-parameters and optimizing them based on the segmenter's response on a separate validation split, we decouple the prompt generation logic from specific training instances. This bi-level feedback loop ensures that the detector learns a prompting strategy that maximizes downstream mask quality on unseen data, effectively bridging the objective gap between geometric localization and segmentation-optimal prompting.

## 2.3 Bi-level optimization

Bi-level optimization (BLO) formulates learning as a nested problem where a lower-level optimization task is constrained by an upper-level objective Liu et al. (2021). This framework has been widely applied to neural architecture search Liu et al. (2018a); Hosseini et al. (2023), hyper-parameter optimization Liu et al. (2022); Kim & Hospedales (2025), and data reweighting Chen et al. (2021); Zhang et al. (2025). In these applications, model parameters are typically optimized in the lower level on a training split, while meta-parameters (e.g. hyper-parameters, architectures, or sample weights) are learned in the upper level on a separate validation split to maximize generalization. Significant progress has been made in developing efficient gradient-based methods for BLO problems. Liu et al. (2018a) introduced a finite difference approximation to estimate upper-level gradients without Hessian computation, while Finn et al. (2017) proposes to compute the gradient updates of meta variables directly with iterative differentiation Grazzi et al. (2020). Recently, Choe et al. (2022) developed a software framework that enables efficient gradient computation across these various approximation schemes. In this work, we leverage these efficient solvers to implement our proposed framework, adapting the BLO paradigm to align object detection with segmentation foundation models.

# 3 Method

## 3.1 Overview of BLO-Inst

Our proposed framework, BLO-Inst, unifies the object detector (YOLO Redmon et al. (2016)) and the segmentation model (SAM Kirillov et al. (2023)) into a unified instance segmentation system, as shown in Figure 1. Let $\Phi$ denote the trainable parameters of the YOLO detector (acting as the prompt generator),

---

**Algorithm 1** Optimization process for BLO-Inst

---

1: **Input:** Detector (YOLO) $\Phi^{(0)}$, Segmenter (SAM) $\Theta^{(0)}$, Training data $\mathcal{D}$
2: **Params:** Learning rates $\alpha, \beta$
3: Split $\mathcal{D}$ into non-overlapping $D_1$ and $D_2$
4: **for** $t = 0$ **to** $T - 1$ **do**
5:     Sample batches $\mathcal{B}_1 \sim D_1$ and $\mathcal{B}_2 \sim D_2$
6:     *// Lower Level: Update Segmenter*
7:     *// Calculate Total Loss, but gradients only flow to $\Theta$*
8:     $L_{\text{low}} = \mathcal{L}_{\text{total}}(\text{YOLO}(\Phi^{(t)}), \text{SAM}(\Theta^{(t)}); \mathcal{B}_1)$
9:     $\Theta^{(t+1)} \leftarrow \Theta^{(t)} - \alpha\nabla_{\Theta}L_{\text{low}}$
10:    *// Upper level: update detector*
11:    *// Compute loss; gradients flow only to $\Phi$*
12:    $L_{\text{upper}} = \mathcal{L}_{\text{total}}(\text{YOLO}(\Phi^{(t)}), \text{SAM}(\Theta^{(t+1)}); \mathcal{B}_2)$
13:    $\Phi^{(t+1)} \leftarrow \Phi^{(t)} - \beta\nabla_{\Phi}L_{\text{upper}}$
14: **end for**

---

and $\Theta$ denote the trainable parameters of the SAM (including PEFT modules like LoRA, and original lightweight modules). Standard approaches typically optimize these models by a summation of losses on the same dataset. However, this often leads to alignment overfitting, where the detector memorizes specific box adjustments for the training examples rather than learning a generalizable prompting policy. To resolve this, we formulate the training as a BLO problem. We partition the training data $\mathcal{D}$ into two disjoint subsets: $D_1$ and $D_2$. The learning process consists of two nested levels: for the lower level, we fix the detector $\Phi$ and fine-tune the segmenter $\Theta$ to adapt to the provided prompts on $D_1$; for the upper level, we update the detector $\Phi$ to generate prompts that minimize the validation loss of the fine-tuned segmenter on $D_2$. By validating the prompt quality on unseen data ($D_2$), we force the detector to learn robust adjustment rules. The two levels of problems share the same form of loss function. The two levels are optimized iteratively until convergence, as shown in Algorithm 1.

**Preliminary.** As mentioned above, BLO-Inst builds upon two foundational architectures: YOLO Redmon et al. (2016) and SAM Kirillov et al. (2023). YOLO is a high-efficiency one-stage object detector that regresses bounding box coordinates and class probabilities directly from input images. We use YOLO as the prompt generator, parameterized by $\Phi$. SAM is a promptable segmentation foundation model comprising a heavy image encoder (ViT), a lightweight prompt encoder, and a mask decoder. It is designed to predict zero-shot masks based on the given prompts (points or boxes). In our framework, we employ SAM as the mask generator, parameterized by $\Theta$ (specifically fine-tuning the decoder by LoRA), to produce high-fidelity masks conditioned on the boxes provided by YOLO.

### 3.2 Bi-level optimization framework

**Lower-level problem (Segmentation adaptation).** In the lower level, the detector $\Phi$ is fixed. It generates bounding boxes, which serve as prompts for the segmentation model. We optimize the segmenter parameters $\Theta$ on $D_1$ to minimize the unified objective $\mathcal{L}_{total}$, which is a weighted sum of four components, consistent with standard YOLO training but augmented with SAM's feedback:

$$\mathcal{L}_{total} = \lambda_1\mathcal{L}_{box} + \lambda_2\mathcal{L}_{obj} + \lambda_3\mathcal{L}_{cls} + \lambda_4\mathcal{L}_{seg} \tag{1}$$

where $\lambda$ terms are hyper-parameters balancing the weight of each component, detailed illustration for objective function can be found in Appendix A. The lower level aims to solve the following optimization problem:

$$\Theta^*(\Phi) = \arg\min_{\Theta} \mathcal{L}_{total}(\Theta, \Phi; D_1) \tag{2}$$

Here, $\Theta^*(\Phi)$ implies that the optimal segmentation parameters $\Theta^*$ are dependent on the detector $\Phi$, as the value of the loss function relies on the quality and characteristics of the prompts generated by $\Phi$.

**Upper-level problem (Prompt alignment).** In the upper level, we evaluate the performance of the fine-tuned segmenter $\Theta^*(\Phi)$ on $D_2$. Our goal is to update the detector parameters $\Phi$ to minimize the same unified objective $\mathcal{L}_{total}$ on validation set, $D_2$, which simulates test-time evaluation. The upper level optimization problem is formulated as:

$$\Phi^* = \arg\min_\Phi \mathcal{L}_{total}(\Theta^*(\Phi), \Phi; D_2) \tag{3}$$

This objective forces the detector to find a solution that satisfies two conditions simultaneously: it must maintain high detection precision (via the detection loss terms) and, more importantly, it must generate generalizable prompts that minimize the validation loss of the segmenter on unseen data ($D_2$) to prevent the alignment-overfitting. Unlike the lower level where $\Phi$ is fixed, here $\Phi$ is the active variable.

**Bi-level optimization framework.** Integrating the aforementioned two optimization problems, we unify them into a cohesive bi-level optimization framework:

$$
\begin{aligned}
\min_\Phi \quad & \mathcal{L}_{total}(\Theta^*(\Phi), \Phi; D_2) \\
\text{s.t.} \quad & \Theta^*(\Phi) = \arg\min_\Theta \mathcal{L}_{total}(\Theta, \Phi; D_1)
\end{aligned}
\tag{4}
$$

In this framework, the two optimization problems are deeply interdependent, structurally functioning as a cross-validation mechanism. The output of the lower level, $\Theta^*(\Phi)$, represents the segmenter's optimal adaptation to the current prompts evaluated on the training split $D_1$. Conversely, the optimization variable in the upper level (the detector $\Phi$) is updated strictly based on the adapted segmenter's validation loss on the disjoint split $D_2$. By explicitly evaluating and optimizing the detector on unseen validation data, this nested structure decouples the prompt generation logic from specific training instances in $D_1$. This prevents alignment overfitting and ensures the learned prompting policy is robust and generalizable to new images.

### 3.3 Optimization algorithm

We employ a gradient-based optimization algorithm to solve the bi-level problem defined in Eq. (4). Since obtaining the exact optimal solution $\Theta^*(\Phi)$ in the lower level is computationally infeasible for every upper-level update, we adopt an efficient approximation strategy inspired by Liu et al. (2018a). As shown in Algorithm 1, instead of fully training the segmenter to convergence at each step, we approximate $\Theta^*(\Phi)$ using a one-step gradient descent update. At iteration $t$, given the current detector $\Phi^{(t)}$, we update the segmenter parameters $\Theta^{(t)}$ on the batch $\mathcal{B}_1$. Then, we use this updated surrogate $\Theta'$ to approximate the optimal segmenter $\Theta^*(\Phi^{(t)})$ for the subsequent upper-level update. To make this computationally feasible, we utilize a one-step gradient descent approximation to estimate the segmenter's optimal adaptation $\Theta^*(\Phi)$. For the detector update in the upper level, we employ a finite difference approximation to handle the implicit Hessian-vector product. This reduces computational overhead while preserving the necessary second-order information for stable alignment. A detailed derivation of this optimization, along with the full differentiable computational path connecting the YOLO outputs to the SAM prompts, is provided in Appendix C.

## 4 Experiments

In this section, we evaluate BLO-Inst across a diverse set of instance segmentation tasks, ranging from general object detection to fine-grained part segmentation and biomedical object detection. We aim to demonstrate that our proposed bi-level optimization framework effectively prevents overfitting when aligning the prompt generator (YOLO) with the segmenter (SAM) to outperform other specialist and automated prompting baselines.

### 4.1 Datasets

We evaluate BLO-Inst on 6 publicly available datasets, categorizing them into general and biomedical object benchmarks to assess domain generalization. For general object detection, we utilize PennFudanPed Wang et al. (2007) for pedestrian segmentation, TransIns SEG (2023) for vehicle and laneline detection under

varied conditions, WheatIns Shehadeh (2024) for dense agricultural object detection, and CarPartIns Pasupa et al. (2022) for fine-grained multi-class segmentation of vehicle components. To assess performance in the biomedical domain, we employ CellCountIns learning (2024), a binary dataset for cell counting in low-contrast microscopy images, and RWCellIns atri (2023), a multi-class benchmark distinguishing red and white blood cells. These datasets vary significantly in scale, density, and complexity, ranging from binary to multi-class tasks, ensuring a robust evaluation across different domains. More details about the datasets can be found in Appendix D. In our method, the training set is further randomly split into two subsets $D_1$ and $D_2$ with equal size. Baseline methods utilize the entire training set without any subdivision.

## 4.2 Experimental settings

**Baselines and metrics.** To comprehensively evaluate the effectiveness of our proposed framework, we compare BLO-Inst against a diverse set of state-of-the-art baselines categorized into two groups: (1) Specialist Instance Segmenters, including the representative two-stage Mask R-CNN He et al. (2017) (abbreviated as "Mask R") and the one-stage box-free SOLO Wang et al. (2021); (2) Automated Prompting Approaches, including SAM-seg Kirillov et al. (2023) with both Mask R-CNN (abbreviated as "SAM+B") and Mask2Former (abbreviated as "SAM+M") variants, which utilize pretrained specialist detectors to provide box prompts to the SAM (all parameters would be further fine-tuned on the target datasets), RSPrompter Chen et al. (2024) with both anchor-based (abbreviated as "RS+Anchor") and query-based (abbreviated as "RS+Query") variants, and USIS Lian et al. (2024), which employ auxiliary networks for prompt generation. Notably, if not specified, the mentioned abbreviations are applied to the following results presentations for both figures and tables. Following standard evaluation protocols, we report the Mean Average Precision (mAP) averaged over IoU thresholds from 0.5 to 0.95. To provide a more granular assessment of detection recall and segmentation mask fidelity, we also report $AP_{50}$ and $AP_{75}$ (detailed definitions are provided in Appendix B).

**Implementation details.** We implement our framework using PyTorch. All experiments are conducted on a single NVIDIA A100 GPU. For the model architecture, we employ YOLOv7 Redmon et al. (2016) as the prompt generator ($\Phi$) and SAM ViT-B Kirillov et al. (2023) initialized with SA-1B weights as the segmenter ($\Theta$). To ensure parameter efficiency, the SAM backbone is frozen, and we fine-tune only the lightweight mask decoder via injected LoRA layers with a rank of $r = 4$. To ensure the detector is adapted to the specific domain before bi-level optimization, the YOLO component is pretrained on the training set of the target dataset for 100 iterations. During the bi-level optimization phase, the model is fine-tuned end-to-end for 20 epochs. We use Stochastic Gradient Descent (SGD) Bottou (2010) for both optimization levels, with learning rates set to $\alpha = 1 \times 10^{-3}$ for the lower level (segmenter update) and $\beta = 1 \times 10^{-3}$ for the upper level (detector update), adjusted via a LambdaLR scheduler. Regarding the unified objective function, the trade-off parameters are set as follows: box regression loss $\lambda_1 = 0.3$, objectness loss $\lambda_2 = 0.7$, classification loss $\lambda_3 = 0.3$, and segmentation loss $\lambda_4 = 0.7$. We assign higher weights to $\lambda_{obj}$ and $\lambda_{seg}$ to prioritize object discovery and final mask fidelity, while the relatively lower weights for $\lambda_{box}$ and $\lambda_{cls}$ leverage SAM's inherent robustness to approximate spatial prompts, reducing the need for pixel-perfect bounding box regression. Following the data partition strategy defined in Section 3, we randomly split the training set into two equal-sized subsets, $D_1$ and $D_2$, to run the bi-level optimization.

## 4.3 Results and analysis

**Single-class general object benchmarks.** We first evaluate BLO-Inst on single-class segmentation tasks using the PennFudanPed and WheatIns datasets. The quantitative results, along with model complexity and training costs, are presented in Table 1. We can see that BLO-Inst achieves the highest mAP on both benchmarks, demonstrating superior alignment between the prompt generator and segmenter. Notably, on the WheatIns dataset, which features dense occlusions, our method improves mAP by over 4.7% compared to the second best baseline (RS+Query). Such superiority mainly results from our bi-level optimization strategy, which can prevent the alignment overfitting and improve the generalization. Beyond accuracy, Table 1 also highlights the parameter efficiency of our approach. While automated prompting methods like USIS and RSPrompter introduce auxiliary networks (up to 100M+ trainable parameters) or require fine-tuning the massive SAM backbone, BLO-Inst achieves state-of-the-art performance with only 38.66M

Table 1: Comparison of BLO-Inst with baselines on single-class general object benchmarks. Metrics reported are Mean AP ($mAP$), $AP_{50}$, and $AP_{75}$ (%). Best results are highlighted in **bold**. The last three columns show the total number of model parameters, the number of trainable parameters (in millions), and the training time (in GPU hours) that calculated on PennFudanPed dataset with NVIDIA A100.

| Method | PennFudanPed | | | WheatIns | | | Total Param(M) | Trainable Param(M) | Train Cost (GPU hours) |
|---|---|---|---|---|---|---|---|---|---|
| | $mAP$ | $AP_{50}$ | $AP_{75}$ | $mAP$ | $AP_{50}$ | $AP_{75}$ | | | |
| Mask R | 46.4 | 81.8 | 49.0 | 48.8 | 81.9 | 64.8 | 41.71 | 41.48 | 0.38 |
| SOLO | 42.3 | 75.7 | 39.7 | 58.8 | 91.5 | 71.3 | 46.23 | 46.01 | 0.43 |
| SAM+B | 54.3 | 80.5 | 63.7 | 59.2 | 87.9 | 79.6 | 111.08 | 111.08 | 1.25 |
| SAM+M | 54.1 | 78.9 | 61.3 | 60.4 | 88.1 | 75.9 | 112.18 | 112.18 | 2.48 |
| RS+Anchor | 41.1 | 68.4 | 45.9 | 62.9 | 87.0 | 78.3 | 117.05 | 117.05 | 0.43 |
| RS+Query | 53.3 | 76.3 | 62.4 | 63.7 | 86.1 | 79.7 | 100.99 | 100.99 | 1.43 |
| USIS | 55.1 | 72.6 | 61.0 | 62.5 | 88.5 | 80.3 | 698.10 | 57.02 | 0.68 |
| **BLO-Inst** | **59.3** | **86.9** | **67.8** | **68.4** | **95.6** | **84.0** | 131.63 | 38.66 | 0.51 |

Table 2: Comparison of BLO-Inst with baselines on multi-class general object benchmarks. Metrics reported are Mean AP ($mAP$), $AP_{50}$, and $AP_{75}$ (%). Best results are highlighted in **bold**.

| Method | TransIns | | | CarPartsIns | | |
|---|---|---|---|---|---|---|
| | $mAP$ | $AP_{50}$ | $AP_{75}$ | $mAP$ | $AP_{50}$ | $AP_{75}$ |
| Mask R | 37.7 | 65.5 | 38.4 | 34.0 | 61.7 | 34.9 |
| SOLO | 47.4 | 78.8 | 49.1 | 49.1 | 77.7 | 51.7 |
| SAM+B | 57.8 | 86.6 | 53.1 | 58.4 | 79.6 | 64.2 |
| SAM+M | 54.9 | 81.5 | 57.0 | 61.7 | 80.8 | 64.0 |
| RS+Anchor | 57.3 | 88.3 | 59.9 | 54.5 | 73.9 | 58.8 |
| RS+Query | 55.7 | 82.5 | 59.2 | 61.1 | 80.6 | 64.7 |
| USIS | 38.8 | 65.5 | 38.9 | 57.9 | 83.5 | 61.1 |
| **BLO-Inst** | **58.8** | **90.2** | **62.2** | **65.2** | **86.5** | **67.2** |

Table 3: Comparison of BLO-Inst with baselines on biomedical benchmarks. Metrics reported are Mean AP ($mAP$), $AP_{50}$, and $AP_{75}$ (%). Best results are highlighted in **bold**.

| Method | CellCountIns | | | RWCellIns | | |
|---|---|---|---|---|---|---|
| | $mAP$ | $AP_{50}$ | $AP_{75}$ | $mAP$ | $AP_{50}$ | $AP_{75}$ |
| Mask R | 42.0 | 74.9 | 41.6 | 63.2 | 90.5 | 76.7 |
| SOLO | 41.6 | 63.7 | 37.4 | 64.7 | 89.5 | 74.6 |
| SAM+B | 50.9 | 82.3 | 60.4 | 76.1 | 91.6 | 88.2 |
| SAM+M | 49.7 | 74.2 | 50.9 | 75.0 | 91.8 | 84.7 |
| RS+Anchor | 49.8 | 78.7 | 55.2 | 76.1 | 92.5 | 88.1 |
| RS+Query | 38.0 | 58.0 | 42.4 | 71.5 | 90.4 | 83.4 |
| USIS | 34.8 | 67.6 | 33.2 | 74.3 | 90.8 | 85.2 |
| **BLO-Inst** | **55.1** | **83.7** | **62.5** | **78.5** | **94.6** | **89.8** |

trainable parameters. This is achieved by freezing the SAM backbone and only updating the lightweight LoRA layers and the YOLO detector. Consequently, our training cost (0.51 GPU hours) is significantly lower or comparable than other SAM-based fine-tuning approaches, validating that our bi-level strategy maintains a high computational efficiency while improving model's performance.

**Multi-class general object benchmarks.** We next evaluate performance on multi-class segmentation tasks using TransIns (Cars vs. LaneLines) and CarPartsIns (Wheels vs. Windows vs. Body, etc.). Results are summarized in Table 2. BLO-Inst consistently outperforms baselines in these complex scenarios. On CarPartsIns, which requires fine-grained discrimination between geometrically distinct components, our method achieves an $AP_{75}$ of 67.2%, significantly surpassing the best automated prompting baseline, RS+Query (64.7%). This indicates that our bi-level optimization effectively teaches the detector to generate class-discriminative prompts, bounding boxes, that not only localize the object but are tightly fitted to trigger the specific semantic mask within SAM's decoder.

**Biomedical benchmarks.** Finally, to assess domain generalization, we report results on the CellCountIns and RWCellIns datasets in Table 3. Despite the significant domain gap between natural images and microscopy, BLO-Inst demonstrates robust adaptability. On the multi-class RWCellIns dataset, our method achieves a remarkable 94.6% $AP_{50}$ and 89.8% $AP_{75}$, outperforming the other automated prompting baselines (which may suffer alignment overfitting) and the specialist Mask R-CNN. This confirms that our method

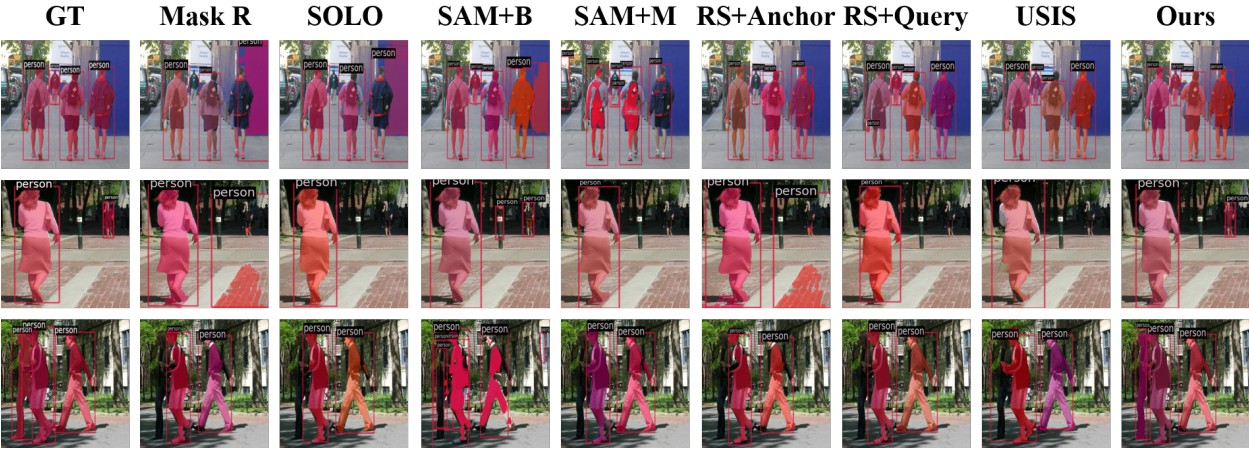

Figure 3: Qualitative comparisons on Natural Scenes (PennFudanPed). Our method (far right) produces sharper masks and handles occlusions more effectively than baselines.

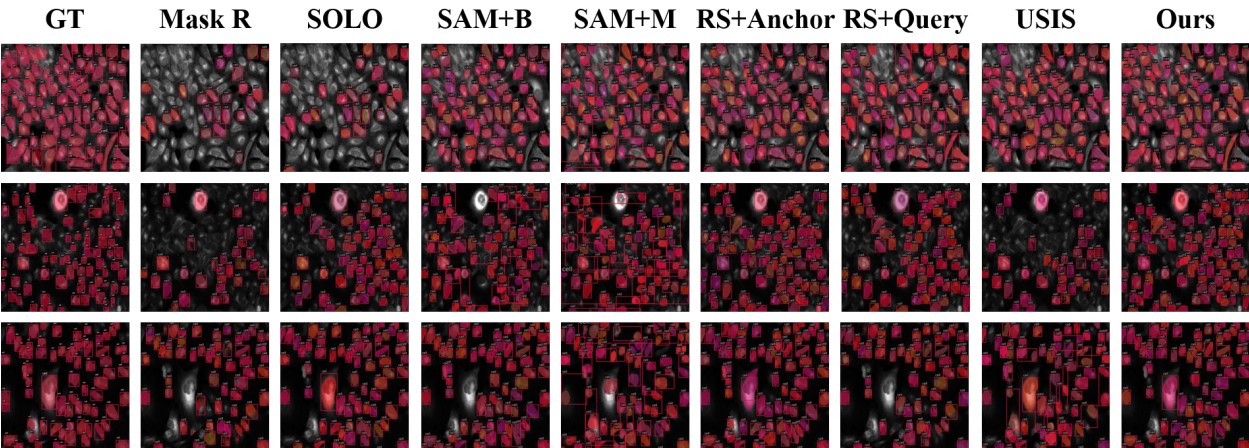

Figure 4: Qualitative Results on Biomedical Imagery (CellCountIns). In this challenging dense microscopy task, BLO-Inst successfully separates tightly packed and overlapping cells, whereas other baselines tend to merge adjacent instances.

successfully bridges the domain gap, aligning the detector's prompting strategy with the geometric properties of biomedical objects to achieve high-fidelity segmentation.

**Performance in Low-Data Regimes.** Our framework is explicitly designed to solve the alignment over-fitting problem that occurs when optimizing a high-capacity foundation model (like SAM) alongside an object detector on specialized, low-data downstream benchmarks. To empirically validate performance in such data-scarce environments, we conducted a data-scaling experiment on the standard MS COCO dataset Lin et al. (2014) by sub-sampling the training split at 1% and 5% configurations. Table 4 summarizes the comparisons between standard single-level optimization and our proposed bi-level optimization strategy.

At a highly restricted 1% training split on COCO, our bi-level framework achieves substantial performance gains of +4.9% mAP, +6.7% $AP_{50}$, and +5.1% $AP_{75}$ over single-level joint training. This explicitly confirms that the nested split-data optimization strategy successfully prevents the detector from learning destructive, overfitted prompt policies. When the training data increases to 5%, the performance gap narrows slightly (+3.3% mAP and +5.7% $AP_{50}$), though our method maintains a commanding lead across all thresholds. This exact trend confirms that our approach is uniquely powerful in data-scarce environments, where overfitting typically degrades the performance of standard joint-training paradigms.

Table 4: Performance comparison on the MS COCO dataset under low-data regimes (1% and 5% training splits). Metrics reported are Mean AP (mAP), $AP_{50}$, and $AP_{75}$ (%). Absolute improvements over the single-level baseline are shown in parentheses.

| Data Split | Strategy | mAP | $AP_{50}$ | $AP_{75}$ |
|---|---|---|---|---|
| **COCO (1% Train)** | Single-level | 22.5 | 40.2 | 19.7 |
| | **Bi-level (Ours)** | **27.4 (+4.9)** | **46.9 (+6.7)** | **24.8 (+5.1)** |
| **COCO (5% Train)** | Single-level | 27.8 | 49.5 | 28.3 |
| | **Bi-level (Ours)** | **31.1 (+3.3)** | **55.2 (+5.7)** | **32.8 (+4.5)** |

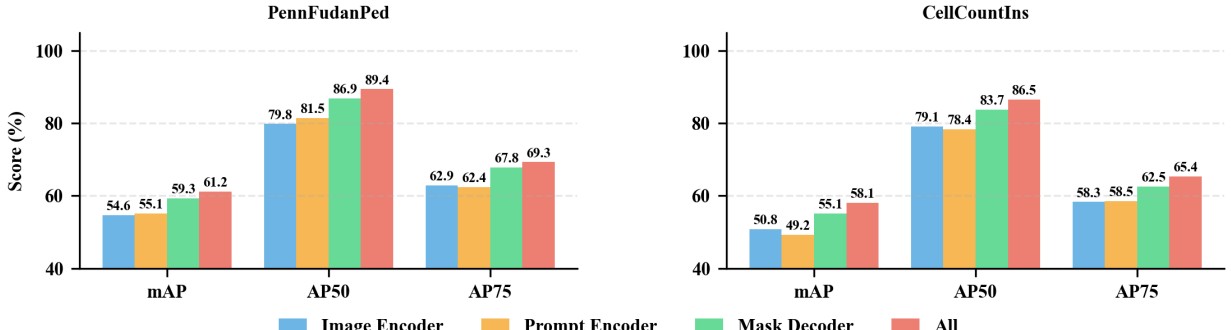

Figure 5: Effect of Trainable Components. Comparison of fine-tuning different modules of SAM on PennFudanPed and CellCountIns datasets. Updating the mask decoder yields the optimal balance between accuracy and parameter efficiency.

**Qualitative analysis.** To visualize the effectiveness of BLO-Inst, we present qualitative comparisons against state-of-the-art baselines in Figure 3 and Figure 4. On the PennFudanPed dataset, traditional fully supervised methods like Mask R-CNN often struggle with precise boundary adherence, while recent automated prompting methods (e.g., USIS) occasionally exhibit prompt misalignment leading to fragmented masks. In contrast, BLO-Inst produces sharp, cohesive masks that accurately delineate instances even under occlusion. Furthermore, in the challenging high-density environment of CellCountIns, baseline prompt-learning approaches (e.g., RSPrompter) frequently suffer from "instance merging," where adjacent cells are grouped into a single mask. Our method successfully separates these tightly packed instances with high fidelity, demonstrating that our bi-level alignment strategy effectively teaches the detector to generate discriminative prompts tailored to the specific segmentation properties of the SAM backbone. Additional qualitative results on traffic, agriculture, and industrial datasets are provided in the Appendix E.

**Robustness and Statistical Significance.** To ensure the reliability of our findings and mitigate the impact of favorable initializations or data splits, we conducted multiple independent runs for our primary benchmarks and key ablations. Specifically, for the PennFudanPed and CellCountIns datasets, we performed three independent training cycles, each utilizing a unique random seed and a fresh random partition of the $D_1$ and $D_2$ subsets. The resulting mean and standard deviation metrics (detailed fully in Appendix F) confirm that the performance gains of BLO-Inst are statistically significant. Furthermore, the low variance across runs demonstrates that our nested feedback loop grounds the detector's updates in the segmenter's actual performance, leading to a highly stable and reliable optimization trajectory compared to standard joint-training baselines.

## 4.4 Ablation studies

To validate the contributions of each component and design choice in BLO-Inst, we conduct ablation studies on the PennFudanPed and CellCountIns datasets. Unless otherwise stated, all ablations use the default settings: fine-tuning the mask decoder by LoRA, using first-order optimization, and a 1:1 data split ratio.

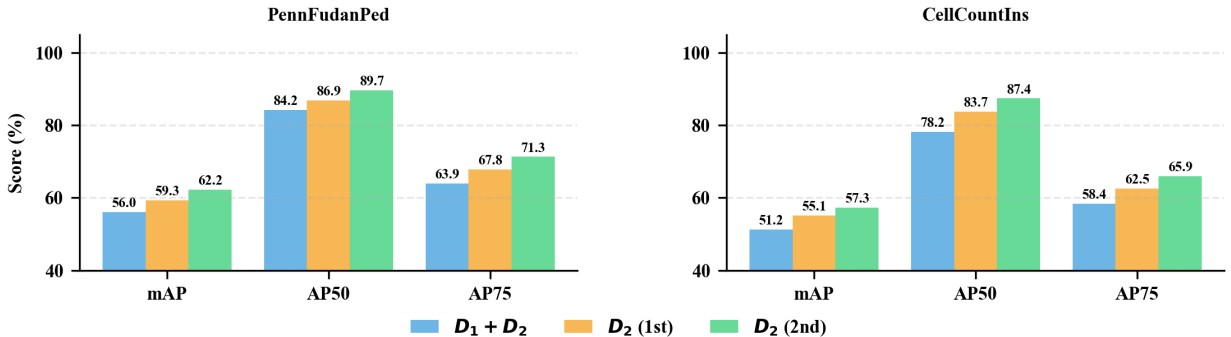

Figure 6: Optimization Strategy Analysis. Performance comparison between single-level training ($D_1 + D_2$) and bi-level optimization. The proposed bi-level strategy prevents overfitting compared to the baseline.

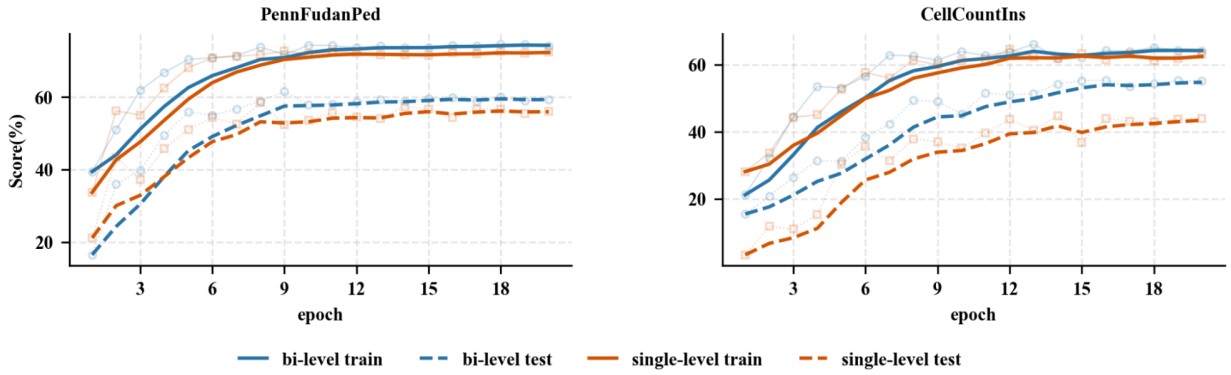

Figure 7: Performance gap analysis demonstrating alignment overfitting on the PennFudanPed and Cell-CountIns datasets. The raw per-epoch Mean AP (faded markers) and EMA-smoothed trends (solid/dashed lines) are shown for both single-level (orange) and bi-level (blue) optimization. Single-level optimization exhibits a widening gap between training and testing performance, indicating overfitting. In contrast, bi-level optimization maintains a significantly narrower gap, demonstrating superior generalization.

**Effectiveness of trainable components.** We first investigate which modules of the SAM should be optimized to adapt to downstream tasks. We compare four settings: updating the image encoder, prompt encoder, mask decoder, or all three components. As illustrated in Figure 5, fine-tuning the heavy image encoder yields suboptimal performance (e.g., 54.6% mAP on PennFudanPed) and incurs high training costs. This is attributed to the fact that the original image encoder, having been pre-trained on the massive SA-1B dataset, already possesses highly robust and generalizable feature extraction capabilities. Consequently, aggressively fine-tuning this backbone on small downstream datasets offers diminishing returns compared to the computational overhead. In contrast, updating the mask decoder alone significantly boosts performance to 59.3%, proving that adapting the segmentation head is crucial for aligning SAM with the prompt generator. While fine-tuning all components yields a marginal further gain (61.2%), it comes at a massive parameter cost. Thus, we select the mask decoder as the optimal trade-off between efficiency and accuracy, as it achieves significantly higher performance than updating the heavy image encoder (59.3% versus 54.6% mAP) while requiring only 38.66M trainable parameters, far fewer than fine-tuning the entire model.

**Impact of optimization strategy.** We analyze the effectiveness of our bi-level optimization strategy by comparing it against a standard single-level baseline ("$D_1 + D_2$"), where the detector and segmenter are trained jointly on the union of datasets. We also compare first-order, "$D_2$ (1st)", with second-order, "$D_2$ (2nd)", bi-level optimization, as shown in Figure 6. The results demonstrate that the bi-level optimization ($D_2$) strategies consistently outperform the single-level baseline (56.0% mAP) for both tasks. This confirms that splitting the data and using the segmenter's validation loss to update the detector prevents overfitting

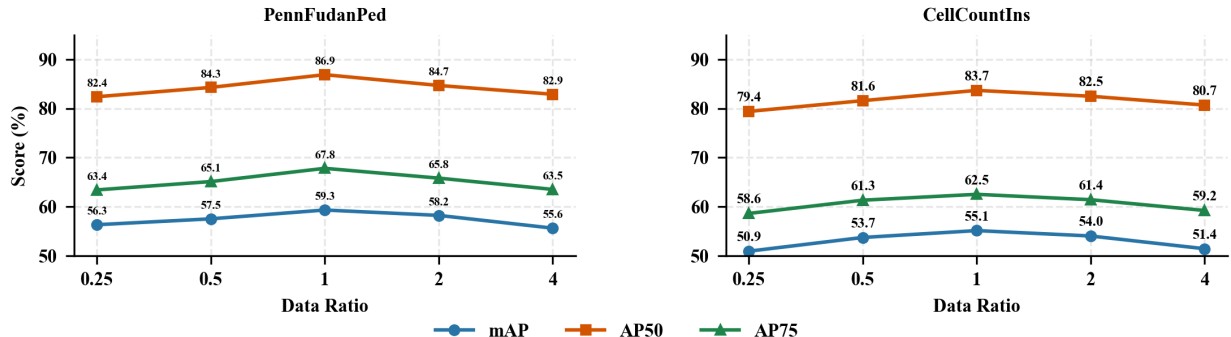

Figure 8: Sensitivity to Data Split Ratio. Mean AP performance across varying ratios ($\gamma$) of subset ($D_1$) to subset ($D_2$). A balanced split ($\gamma = 1$) proves most effective for convergence.

and improves generalization. Although the second-order approximation achieves the highest performance (62.2% mAP) on pedestrian segmentation task, it requires significantly more computational resources. Therefore, we utilize the first-order optimization as our default setting to reduce training cost, while maintaining competitive accuracy.

**Direct evidence of alignment overfitting.** To provide explicit empirical evidence of the alignment overfitting mechanism, we analyze the training and testing performance gaps throughout the optimization process. Figure 7 illustrates the Mean AP scores on both the training and testing splits for the PennFudanPed and CellCountIns datasets. In the standard single-level optimization baseline (orange curves), we observe a significantly larger and growing gap between the training and testing performance. This indicates that while the detector successfully minimizes the loss on training samples, it fails to generalize these learned prompt adjustments to unseen data.

In contrast, our bi-level optimization strategy (blue curves) maintains a much narrower performance gap throughout the entire training trajectory. By explicitly optimizing the detector on the disjoint validation split ($D_2$), the bi-level framework forces the model to learn robust, generalizable prompting rules rather than memorizing sample-specific coordinate offsets. To ensure visual clarity, Figure 7 displays both the raw per-epoch values and an Exponential Moving Average (EMA) smoothed trend line, confirming that the bi-level approach consistently prevents overfitting and achieves superior convergence on the test set.

**Sensitivity to data split ratio.** Finally, we explore the impact of the data split ratio between the subset $D_1$ (for $\Theta$) and subset $D_2$ (for $\Phi$). We test ratios of $\gamma = |D_1|/|D_2|$ ranging from 0.25 to 4. As shown in Figure 8, a balanced split ($\gamma = 1$) yields the best performance (59.3% mAP on PennFudanPed). Skewing the data too heavily towards the subset for segmenter updating (Ratio 4) starves the optimization process of detector, leading to poor prompt generation (dropping to 55.6%). Conversely, skewing towards the subset for detector updating (Ratio 0.25) prevents the segmenter from learning adequate domain adaptations. A balanced ratio provides sufficient data for both levels of the optimization to converge effectively.

**End-to-End vs. Separate training.** We validate the necessity of our unified training pipeline in Figure 9. In the "Separate" setting, we first train the YOLO detector to convergence and then freeze it to train the SAM segmenter. In the "End-to-End" setting, both are updated simultaneously by the bi-level optimization framework. The results reveal that end-to-end training yields significant improvements (+6.2% mAP on PennFudanPed). In the separate approach, the detector learns to optimize bounding boxes solely, which does not necessarily correlate with the optimal prompts for SAM. Our end-to-end approach allows the detector to learn how to prompt SAM to maximize mask quality, creating a synergistic feedback loop.

**Component-wise Decoupling Baselines.** To further decouple the performance gains of our framework, we evaluate a suite of baseline configurations across both the PennFudanPed and CellCountIns benchmarks, summarized in Table 5. Specifically, we compare our default bi-level configuration against: completely frozen pre-trained models (SAM + YOLO Zero-shot); training only the detector (Frozen SAM + Finetuned YOLO);

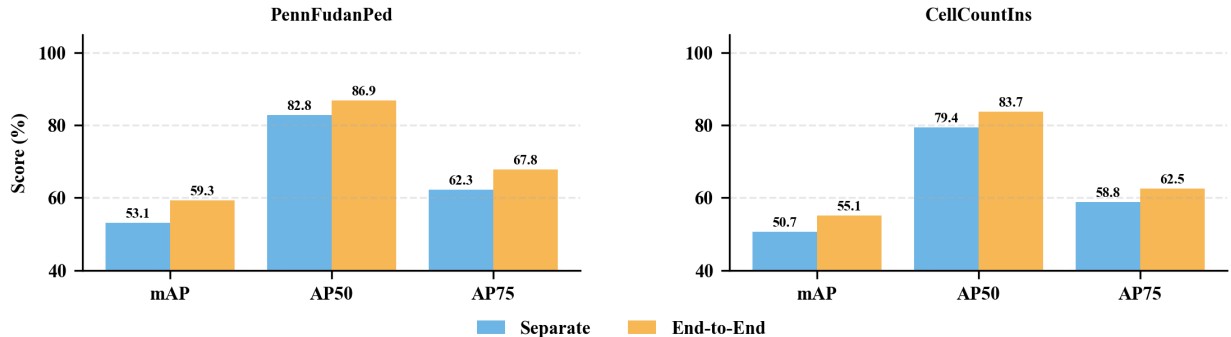

Figure 9: End-to-End vs. Separate Training. Jointly optimizing the detector and segmenter via the bi-level feedback loop (End-to-End) consistently outperforms the sequential (Separate) training pipeline.

Table 5: Decoupled baseline comparisons on PennFudanPed and CellCountIns. This evaluates the impact of freezing versus fine-tuning individual components within the pipeline.

| Dataset | Configuration | mAP | $AP_{50}$ | $AP_{75}$ |
|---|---|---|---|---|
| **PennFudanPed** | SAM + YOLO (Zero-shot) | 41.8 | 85.3 | 33.1 |
| | Frozen SAM + Finetuned YOLO | 58.9 | 83.3 | 62.3 |
| | Finetuned SAM + Frozen YOLO | 57.4 | 84.1 | 62.6 |
| | **Ours (BLO-Inst)** | **59.3** | **86.9** | **67.8** |
| | Full-FT SAM Bi-level | 60.8 | 87.6 | 68.2 |
| **CellCountIns** | SAM + YOLO (Zero-shot) | 6.7 | 12.1 | 6.5 |
| | Frozen SAM + Finetuned YOLO | 53.1 | 82.5 | 61.1 |
| | Finetuned SAM + Frozen YOLO | 54.1 | 82.7 | 61.7 |
| | **Ours (BLO-Inst)** | **55.4** | **83.6** | **62.9** |
| | Full-FT SAM Bi-level | 56.3 | 84.2 | 64.1 |

updating only the segmenter (Finetuned SAM + Frozen YOLO); and expanding the bi-level framework to fine-tune all SAM components rather than just utilizing LoRA (Full-FT SAM Bi-level).

The poor performance of the vanilla zero-shot combination, especially on the challenging biomedical Cell-CountIns benchmark (6.7% mAP), highlights that pre-trained object detectors do not inherently produce prompts tailored to SAM's segmentation tendencies. Furthermore, simply fine-tuning the detector while freezing SAM yields sub-optimal performance compared to our coupled framework. Finally, moving from our default configuration to a Full-FT SAM Bi-level strategy yields a minor performance bump (+1.5% mAP on PennFudanPed, +0.9% mAP on CellCountIns). However, this requires calculating second-order gradients across the massive SAM image encoder, which spikes both computational and GPU memory overhead.

## 5 Conclusion

In this paper, we presented BLO-Inst, a parameter-efficient instance segmentation framework that aligns the SAM with a pretrained detector through bi-level optimization. By formulating the prompt generator as a learnable "hyper-parameter" optimized to maximize SAM's validation performance, we establish a cooperative feedback loop that bridges the objective gap between geometric localization and segmentation map prediction. Extensive experiments across six diverse benchmarks, including challenging biomedical microscopy, demonstrate that BLO-Inst significantly outperforms both fully supervised specialist models and state-of-the-art automated prompting approaches. These findings validate bi-level optimization as a robust paradigm for adapting foundation models to complex downstream tasks, offering a promising direction for future research in automated prompt learning.

## Acknowledgments

We acknowledge funding support from NSF IIS2405974, NSF IIS2339216, and NIH R35GM157217.

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

## A Objective function

For both the lower-level and upper-level optimization steps, we utilize a unified multi-task objective function $\mathcal{L}_{total}$. This objective combines the detection losses required to maintain YOLO's localization capabilities with the segmentation loss required for SAM's mask generation. The total loss is defined as:

$$\mathcal{L}_{total} = \lambda_1 \mathcal{L}_{box} + \lambda_2 \mathcal{L}_{obj} + \lambda_3 \mathcal{L}_{cls} + \lambda_4 \mathcal{L}_{seg} \tag{5}$$

where $\lambda$ terms are hyper-parameters governing the weight of each component. The individual loss components are calculated as follows:

- **Box Regression Loss ($\mathcal{L}_{box}$):** We employ the Complete Intersection over Union (CIoU) loss Zheng et al. (2020) to penalize bounding box regression errors. For a predicted bounding box $b_{pred}$ and a ground truth target box $b_{gt}$, the loss is defined as:

$$\mathcal{L}_{box} = 1 - \text{CIoU}(b_{pred}, b_{gt}) \tag{6}$$

- **Objectness and Classification Loss ($\mathcal{L}_{obj}, \mathcal{L}_{cls}$):** We use Binary Cross-Entropy (BCE) supervised by Focal Loss Lin et al. (2017) to address class imbalance between foreground and background objects. The loss formulation is:

$$\mathcal{L}_{obj/cls} = -\alpha(1 - p_t)^\gamma \log(p_t) \tag{7}$$

where $p_t$ is the predicted probability of the target class, and $\alpha$ and $\gamma$ are hyper-parameters that modulate the loss contribution of hard and easy training examples.

- **Segmentation Loss ($\mathcal{L}_{seg}$):** This serves as the critical link between the detection and segmentation modules. We compute the pixel-wise Binary Cross-Entropy between the predicted mask $M_{pred}$ and the ground truth mask $M_{gt}$. To focus the loss strictly on the relevant instance, we crop the loss calculation to the region defined by the bounding box $b$:

$$\mathcal{L}_{seg} = \frac{1}{|b|} \sum_{(i,j) \in b} \text{BCE}(M_{pred}^{(i,j)}, M_{gt}^{(i,j)}) \tag{8}$$

The deployment of this unified multi-task objective confers a critical dual advantage. First, the inclusion of standard detection losses ($\mathcal{L}_{box}, \mathcal{L}_{obj}, \mathcal{L}_{cls}$) acts as a geometric anchor, preventing the detector from degenerating. Without these constraints, the detector might learn to exploit the segmenter's biases, generating trivial or physically implausible prompts (e.g., encompassing the entire image) solely to minimize segmentation error, thereby losing its ability to distinguish distinct instances. Second, the integration of the segmentation loss ($\mathcal{L}_{seg}$) transforms the optimization landscape, forcing the detector to seek a solution that is segmentation-optimal for both tasks. Unlike standard multi-task learning, where losses are simply summed during a single training phase, our bi-level formulation ensures that the detection parameters are updated specifically to minimize the validation risk of the segmenter, effectively encoding downstream mask fidelity into the bounding box regression coordinates.

## B Evaluation Metrics

To provide a comprehensive assessment of our model's performance, we report instance segmentation results using standard Average Precision (AP) metrics. Throughout the experiments (e.g., Table 1, Table 2, and Table 3), we report the Mean Average Precision (mAP) averaged over multiple Intersection over Union (IoU) thresholds (from 0.50 to 0.95 with a step size of 0.05).

To offer a more granular evaluation of localization and mask fidelity, we also report the following specific threshold metrics:

- $AP_{50}$: This metric represents the Average Precision calculated at a relatively lenient IoU threshold of 0.50. It primarily evaluates the model's fundamental ability to successfully discover and localize objects.

- $AP_{75}$: This metric represents the Average Precision calculated at a stricter IoU threshold of 0.75. It specifically assesses the fine-grained accuracy and tight boundary adherence of the predicted segmentation masks.

## C  Detailed optimization algorithm

In this section, we offer a detailed description of the optimization algorithm of BLO-Inst. We employ a gradient-based optimization algorithm to tackle the problem outlined in Eq. (4). Drawing inspiration from Liu et al. (2018a), we approximately update $\Theta^*(\Phi)$ via one-step gradient descent in the lower-level optimization. Then we plug the approximate $\Theta^*(\Phi)$ into the learning process of the detector in the upper level and update $\Phi$ via one-step gradient descent. By using the one-step gradient descent updates for the bi-level optimization framework, we reduce the computational complexity. The detailed derivation of the update is as follows.

**Inner loop adaptation (Lower level).** In the inner loop, we aim to estimate how the segmenter $\Theta$ adapts to the prompts generated by the current detector $\Phi^{(t)}$. Instead of a full training cycle, we compute a surrogate parameter $\Theta^{(t+1)}$ by performing a single gradient descent step on the support set $D_1$. Given the current state $\Theta^{(t)}$ and learning rate $\alpha$, this virtual update is defined as:

$$\Theta^{(t+1)} = \Theta^{(t)} - \alpha \nabla_\Theta \mathcal{L}_{total}(\Theta^{(t)}, \Phi^{(t)}; D_1) \tag{9}$$

**Outer loop alignment (Upper level).** In the outer loop, we optimize the detector $\Phi$ by minimizing the validation loss on $D_2$, conditioned on the adapted segmenter $\Theta^{(t+1)}$. This formulation effectively forces the detector to anticipate the segmenter's reaction to its prompts. The detector update, with learning rate $\beta$, is given by:

$$\Phi^{(t+1)} = \Phi^{(t)} - \beta \nabla_\Phi \mathcal{L}_{total}(\Theta^*(\Phi), \Phi^{(t)}; D_2) \tag{10}$$

To capture the dependency of the optimal segmenter $\Theta^*$ on $\Phi$, we employ the unrolled differentiation method Liu et al. (2018a). By substituting the one-step approximation from Eq. (9) into the upper-level objective, the gradient with respect to $\Phi$ can be approximated as:

$$\nabla_\Phi \mathcal{L}_{total}(\Theta^*(\Phi), \Phi) \approx \nabla_\Phi \mathcal{L}_{total}(\Theta - \alpha \nabla_\Theta \mathcal{L}_{total}(\Theta, \Phi), \Phi) \tag{11}$$

Applying the chain rule to this unrolled objective reveals that the total gradient consists of a direct gradient term and an implicit Hessian-vector product term:

$$\nabla_\Phi \mathcal{L}_{total}(\Theta - \alpha \nabla_\Theta \mathcal{L}_{total}, \Phi) = \nabla_\Phi \mathcal{L}_{D_2}(\Theta^*, \Phi) - \alpha \nabla^2_{\Phi,\Theta} \mathcal{L}_{D_1}(\Theta, \Phi) \cdot \nabla_\Theta \mathcal{L}_{D_2}(\Theta^*, \Phi) \tag{12}$$

where $\mathcal{L}{D_1}$ and $\mathcal{L}_{D_2}$ denote the losses computed on the support and query sets, respectively.

**Hessian approximation.** Computing the mixed second-derivative matrix $\nabla^2_{\Phi,\Theta}$ directly is computationally expensive. To circumvent this, we approximate the Hessian-vector product in the second term of Eq. (12) using a finite difference method. This reduces the complexity to just two forward passes:

$$\nabla_\Phi \mathcal{L}_{D_2}(\Theta^*, \Phi) - \alpha \nabla^2_{\Phi,\Theta} \mathcal{L}_{D_1}(\Theta, \Phi) \cdot \nabla_\Theta \mathcal{L}_{D_2}(\Theta^*, \Phi) \approx \frac{\nabla_\Phi \mathcal{L}_{D_1}(\Theta^+, \Phi) - \nabla_\Phi \mathcal{L}_{D_1}(\Theta^-, \Phi)}{2\epsilon} \tag{13}$$

Here, $\Theta^{\pm} = \Theta \pm \epsilon \nabla_{\Theta^*} \mathcal{L}_{D_2}(\Theta^*, \Phi)$ represents the segmenter weights perturbed in the direction of the upper-level gradient, and $\epsilon$ is a small scalar (e.g., $\epsilon = 0.01/\|\nabla_{\Theta^*} \mathcal{L}_{D_2}(\Theta^*, \Phi)\|_2$).

**Optimization order.** The parameter $\alpha$ acts as a switch between first-order and second-order optimization. Setting $\alpha = 0$ yields a first-order approximation (ignoring the Hessian term), which significantly reduces computational cost. For our primary experiments, we utilize this first-order optimization for efficiency and stability, but we provide an ablation analysis of the full second-order scheme in the experimental section.

### C.1 Computational Path and Differentiability

To clarify how the two models are aligned end-to-end, we detail the full computational chain and gradient flow from the YOLO outputs to the SAM prompts and back to the detector parameters.

**Forward Path (Detector to Prompt):** The YOLO detector predicts bounding box coordinates as continuous tensors, denoted as $\Phi_{box} \in \mathbb{R}^4$. These coordinates are passed directly into SAM's Prompt Encoder. The encoder utilizes positional encodings to represent the box geometry in high-dimensional space. Because this transformation relies on standard matrix operations, this stage is fully differentiable.

**Forward Path (Prompt to Mask):** The encoded prompt is then fed into SAM's Mask Decoder, which we have modified with learnable LoRA layers. Guided by the prompt, the decoder predicts the final segmentation mask $M_{pred}$. This prediction is compared against the ground truth mask $M_{gt}$ to compute the segmentation loss $\mathcal{L}_{seg}$.

**Backward Path (Gradient Flow):** During the upper-level optimization step, we must update the detector based on the segmenter's performance. Gradients from $\mathcal{L}_{seg}$ flow backward through the Mask Decoder and Prompt Encoder directly to the bounding box coordinates $\Phi_{box}$. Since these coordinates are the direct output of the YOLO regression head, the gradient continues to flow back to update the core detector parameters $\Phi$.

**Handling Non-Differentiable Modules:** A common challenge in detection pipelines is the presence of non-differentiable components, such as Non-Maximum Suppression (NMS). In our framework, we address this by using the discrete selection logic strictly for filtering. During training, we select the top-scoring boxes to serve as prompts. The gradients then flow *only* through the continuous coordinates of those selected boxes back to the regression head. This approach is consistent with standard object detection training (e.g., YOLO, Faster R-CNN), where gradients flow smoothly through the box coordinates, bypassing the non-differentiable discrete selection steps.

## D Datasets

To evaluate the effectiveness and generalization capability of BLO-Inst across diverse domains, we conduct comprehensive experiments on six datasets that vary significantly in object density, scale, and class complexity. These datasets are categorized into general object benchmarks and biomedical benchmarks, ensuring a robust assessment of the model's adaptability. The statistics of the datasets are listed in Table 6.

Table 6: Number of data examples in different tasks

| Dataset | PennFudanPed | TransIns | WheatIns | CarPartsIns | CellCountIns | RWCellIns |
|---|---|---|---|---|---|---|
| Train size | 74 | 135 | 392 | 400 | 219 | 126 |
| Test size | 96 | 59 | 170 | 100 | 32 | 212 |

In the realm of general object segmentation, we utilize three distinct binary benchmarks. First, the PennFudanPed [1] dataset serves as a standard benchmark for pedestrian detection and segmentation. It represents a binary task (Pedestrian vs. Background) that rigorously tests the model's ability to handle articulated deformations and frequent occlusions typical of urban environments. Second, we employ the TransIns [2] dataset, which focuses on vehicle detection within transportation surveillance contexts. This benchmark evaluates the model's performance on rigid objects under varying lighting and traffic conditions. Third, we include the WheatIns [3] (global wheat detection) dataset, an agricultural benchmark characterized by ex-

---

[1] https://www.cis.upenn.edu/~jshi/ped_html/

[2] https://universe.roboflow.com/seg-ohfrw/trans-y9kyy/dataset/1

[3] https://universe.roboflow.com/albara-shehadeh-o8han/test-oaige

tremely high object density and severe overlaps. This dataset challenges the model to effectively distinguish crowded instances where standard detectors often fail due to aggressive suppression.

To assess fine-grained semantic capabilities, we utilize the CarPartsIns [4] dataset. Unlike the previous binary tasks, this is a multi-class benchmark requiring the segmentation of specific vehicle components such as wheels, lights, windows, and the car body. This task is particularly significant as it tests the model's capacity for hierarchical semantic differentiation, requiring it to generate distinct prompts for sub-components that possess vastly different aspect ratios and geometric shapes within a single object category.

Finally, we evaluate domain generalization using two biomedical microscopy datasets. CellCountIns [5] is a binary benchmark designed for cell counting and segmentation. It features small, dense, and visually similar instances with low contrast against the background, highlighting the effectiveness of our bi-level fine-tuning in bridging the domain gap between natural scenes and microscopy. Complementing this is the RWCellIns [6] dataset, a multi-class biomedical benchmark that distinguishes between red and white blood cells. This task introduces the challenge of biological class distinction, testing whether the detector can learn to generate class-discriminative prompts to separate biological structures that share similar circular geometries but differ in texture and size.

## E  Additional qualitative results

To further demonstrate the versatility of BLO-Inst, as shown in Figure 10, Figure 11, Figure 13, and Figure 12, we provide qualitative comparisons on four additional benchmarks covering traffic scenes, agriculture, and industrial inspection.

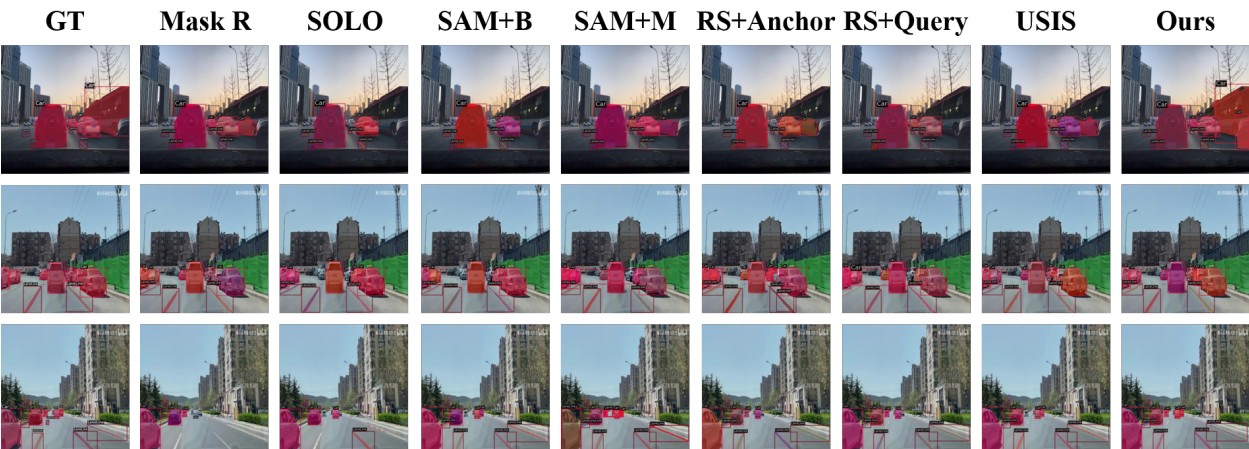

Figure 10: Traffic Scenes (TransIns). Segmentation of vehicles in complex lighting conditions.

## F  Robustness Over Multiple Seeds and Splits

Given the relatively small size of certain downstream datasets and the introduction of the $D_1/D_2$ data split in our bi-level optimization framework, we rigorously evaluated the robustness of BLO-Inst. We report the Mean Average Precision (mAP), $AP_{50}$, and $AP_{75}$ along with their standard deviations across three independent runs. Each run utilized a different random seed and a uniquely randomized $D_1/D_2$ data split.

Table 7 presents the robustness of our main results compared to state-of-the-art baselines on both a natural image dataset (PennFudanPed) and a biomedical dataset (CellCountIns). Table 8 details the variance across our key ablation studies, confirming that our design choices regarding trainable components and optimization strategies yield consistently stable improvements.

---

[4] https://universe.roboflow.com/kmitl-yjl9y/car-parts-segmantation
[5] https://universe.roboflow.com/machine-learning-potko/cell-count-m2j3t/dataset/3
[6] https://universe.roboflow.com/atri-gly8b/cell-p3pcx

Table 7: Main results with mean and standard deviation ($\pm$) across three independent random seeds and data splits.

| Method | PennFudanPed | | | Cell Count | | |
|---|---|---|---|---|---|---|
| | mAP | AP50 | AP75 | mAP | AP50 | AP75 |
| Mask R-CNN | $46.40 \pm 0.31$ | $81.83 \pm 0.45$ | $48.97 \pm 0.53$ | $42.03 \pm 0.46$ | $74.93 \pm 0.84$ | $41.70 \pm 0.74$ |
| SOLO | $42.23 \pm 0.40$ | $75.70 \pm 0.40$ | $39.73 \pm 0.55$ | $41.57 \pm 0.64$ | $63.73 \pm 0.74$ | $37.47 \pm 0.82$ |
| SAM-seg (Mask R) | $54.37 \pm 0.21$ | $80.60 \pm 0.46$ | $63.83 \pm 0.42$ | $50.90 \pm 0.61$ | $82.37 \pm 0.81$ | $60.50 \pm 0.84$ |
| SAM-seg (Mask2F) | $54.07 \pm 0.35$ | $78.97 \pm 0.40$ | $61.33 \pm 0.55$ | $49.70 \pm 0.71$ | $74.30 \pm 0.96$ | $51.00 \pm 0.94$ |
| RSPrompter (anchor) | $41.13 \pm 0.35$ | $68.57 \pm 0.47$ | $46.00 \pm 0.47$ | $49.83 \pm 0.76$ | $78.77 \pm 0.89$ | $55.30 \pm 1.07$ |
| RSPrompter (query) | $53.40 \pm 0.36$ | $76.37 \pm 0.40$ | $62.50 \pm 0.46$ | $38.03 \pm 0.74$ | $58.07 \pm 1.02$ | $42.50 \pm 1.06$ |
| USIS | $55.07 \pm 0.25$ | $72.53 \pm 0.38$ | $60.97 \pm 0.45$ | $34.87 \pm 0.49$ | $67.67 \pm 0.92$ | $33.27 \pm 0.82$ |
| **Ours** | $59.37 \pm 0.21$ | $86.93 \pm 0.35$ | $67.83 \pm 0.45$ | $55.20 \pm 0.43$ | $83.77 \pm 0.59$ | $62.60 \pm 0.77$ |

Table 8: Ablation studies with mean and standard deviation ($\pm$) across three independent random seeds and data splits.

| Setting | PennFudanPed | | | Cell Count | | |
|---|---|---|---|---|---|---|
| | mAP | AP50 | AP75 | mAP | AP50 | AP75 |
| *Model Component* | | | | | | |
| Image encoder | $54.80 \pm 0.26$ | $80.17 \pm 0.36$ | $63.13 \pm 0.49$ | $50.77 \pm 0.55$ | $79.43 \pm 0.67$ | $58.50 \pm 0.92$ |
| Prompt encoder | $55.50 \pm 0.36$ | $81.72 \pm 0.40$ | $62.43 \pm 0.55$ | $49.53 \pm 0.49$ | $78.83 \pm 0.58$ | $58.47 \pm 0.85$ |
| Mask decoder | $59.37 \pm 0.21$ | $86.93 \pm 0.35$ | $67.83 \pm 0.45$ | $55.20 \pm 0.43$ | $83.77 \pm 0.59$ | $62.60 \pm 0.77$ |
| All | $61.57 \pm 0.32$ | $89.20 \pm 0.44$ | $69.17 \pm 0.51$ | $58.57 \pm 0.50$ | $86.77 \pm 0.64$ | $65.13 \pm 0.83$ |
| *Detector Optimization Strategy* | | | | | | |
| $D_1 + D_2$ | $56.23 \pm 0.59$ | $84.03 \pm 0.57$ | $64.23 \pm 0.42$ | $51.17 \pm 0.56$ | $78.13 \pm 0.31$ | $59.07 \pm 0.65$ |
| $D_2$ (1st order) | $59.37 \pm 0.21$ | $86.93 \pm 0.35$ | $67.83 \pm 0.45$ | $55.20 \pm 0.43$ | $83.77 \pm 0.59$ | $62.60 \pm 0.77$ |
| $D_2$ (2nd order) | $62.17 \pm 0.45$ | $89.33 \pm 0.47$ | $71.83 \pm 0.47$ | $57.20 \pm 0.46$ | $87.67 \pm 0.38$ | $65.70 \pm 0.62$ |
| *End-to-End Training* | | | | | | |
| Separate | $53.17 \pm 0.31$ | $82.87 \pm 0.21$ | $62.40 \pm 0.56$ | $50.87 \pm 0.67$ | $79.57 \pm 0.76$ | $58.97 \pm 0.86$ |
| End-to-End | $59.37 \pm 0.21$ | $86.93 \pm 0.35$ | $67.83 \pm 0.45$ | $55.20 \pm 0.43$ | $83.77 \pm 0.59$ | $62.60 \pm 0.77$ |

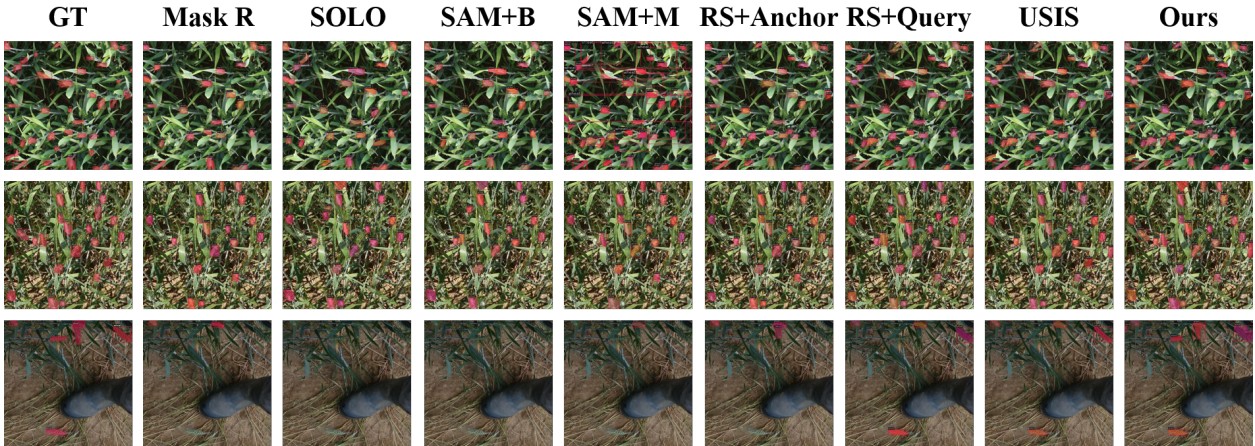

Figure 11: Agriculture (Wheat). Dense instance segmentation of wheat heads in field imagery.

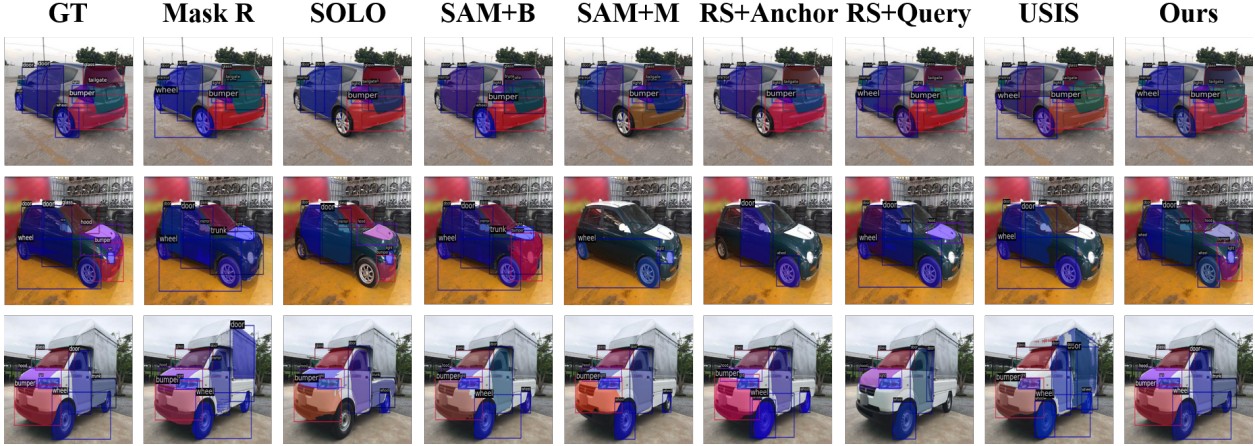

Figure 12: Industrial Inspection (CarParts). Fine-grained segmentation of vehicle components.

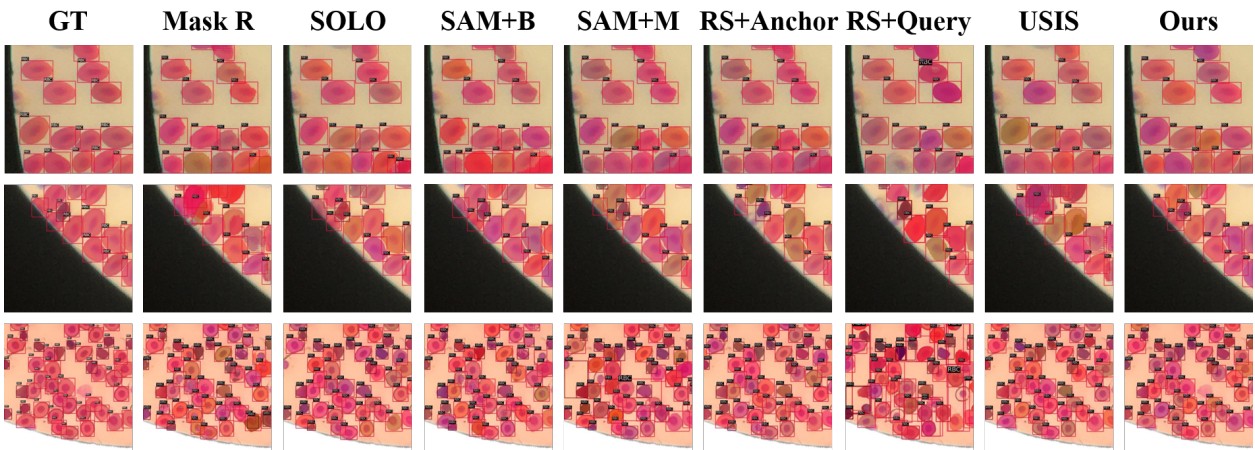

Figure 13: Medical Hematology (RW-Cell). Segmentation of red and white blood cells.

