# OpenReview forum: "Learning to Prompt for Generalizable Instance Segmentation via Bi-Level Optimization"
_TMLR — Accepted by TMLR_

### Review · Reviewer_6Qau · 2026-05-04

**Summary Of Contributions:**

The authors propose using Bi-level optimization, a general optimization technique, for image segmentation. The idea is simple, but it results in meaningful improvements over the state of art.

The authors provide  comparison with previously suggested methods. Selected sample images support authors' contention that bi-level optimization results in better image segmentation. Summary statistics over multiple metrics show consistent improvement.

**Audience:**

Yes

**Audience Explanation:**

Image segmentation is an important and active area of deep learning applications. It has uses in diverse areas from autonomous driving cars to automated processing of medical data.

**Broader Impact Concerns:**

I do not see any concerns.

**Claims And Evidence:**

Yes

**Claims Explanation:**

The authors provide results for 6 diverse and challenging datasets used for image segmentation. I was particularly impressed by good results on blood cells segmentation task.

The authors provided a link to anonymized source code repository for their experiments.

**Requested Changes:**

Critical changes.

In the introduction there is  a nonsensical sentence: "In contrast, the emergence of foundation models has fundamentally shifted this landscape Benigmim et al. (2024); Zhou et al. (2025) for their enrich prior knowledge.".

On page 2 the abbreviation MLP has not been defined.

Equation (1) is not presented well. There are no illustrations in Appendix A that were promised in the sentence following Eq (1). Even after looking at Appendix A, I am unsure about some precise definition of some terms of the loss. References to the papers proposing each term must be included, and ideally the loss function should be better described.

Metrics $AP_{50}$ and $AP_{75}$ used on page 7 have not been defined.

pg. 7 "Implementary" is not a proper English word.

The sentence "We implement our framework using PyTorch, all the experiments are conducted on one NVIDIA A100GPU." should be broken 2. The word "one" could be replaced by "single".

Such superiority mainly "gains" ... -> doesn't sound right, maybe use the verb "results"

pg. 10 The sentence "Thus, we select the mask encoder as the optimal trade-off between efficiency and accuracy" does not make sense to me.


Strenghtening work

I don't like the analogy of hyper-parameters as for instance mentioned in the last sentence at the bottom of page 3. Bi-level optimization is a more general concept, although there are some similarities.

There are numerous repetitions in the paper. For instance datasets $D_1$ and $D_2$ are introduced both on page 3 and page 4.

---

> ### Author Response · Authors · 2026-05-04
> **Revision and Response**
>
> We would like to express our sincere gratitude to the editor and the reviewer for your constructive feedback and the time invested in evaluating our work. Your insights have been instrumental in refining the technical clarity and presentation of our manuscript. The following are the detailed point-to-point responses to the issues raised during the review process.
>
> ### __Response to Critical Changes__
>
> ### 1. Grammatical and Stylistic Corrections
>
> - __Foundation Models Sentence__: We apologize for the phrasing error. We will revise the sentence to: "In contrast, the emergence of foundation models has fundamentally shifted this landscape by leveraging their extensive prior knowledge."
>
> - __"MLP" Definition__: We will update the text on page 2 to define MLP as "Multi-Layer Perceptron (MLP)" upon its first mention.
>
> - __"Implementary" & GPU Sentence__: The term "Implementary" will be changed to "Implementation details". The hardware sentence will be split and revised: "We implement our framework using PyTorch. All experiments are conducted on a single NVIDIA A100 GPU."
>
> - __"Gains" vs. "Results"__: We will replace "mainly gains from" with "mainly results from" to improve the flow and clarity of the results section.
>
> ### 2. Detail Clarifications
>
> - __Loss Function Definition__: We will revise Appendix A to include precise mathematical definitions and citations for each term in $\mathcal{L}_{total}$, including: Complete Intersection over Union (CIoU) loss; Binary Cross-Entropy with Focal Loss; Pixel-wise Binary Cross-Entropy, cropped to the bounding box region.
>
> - __Metric Clarification__:  We will add explicit definitions for these metrics ($AP_{50}$ and $AP_{75}$) in the Appendix, clarifying that they represent Average Precision at IoU thresholds of 0.50 and 0.75, respectively.
>
> - __Mask Decoder Trade-off__: The reviewer found the statement on page 10 confusing. We will further clarify that updating only the mask decoder provides the "optimal trade-off" because it achieves significantly higher performance than updating the image encoder (59.3% vs 54.6% mAP) while requiring only 38.66M trainable parameters, which is far fewer than fine-tuning the entire model.
>
> ### __Strengthening the Work__
>
> ### 1. The Hyper-parameter Analogy
>
> We acknowledge the reviewer's point that bi-level optimization (BLO) is a broader concept than hyper-parameter tuning. However, we used the analogy to describe how the detector's bounding boxes act as "dynamic hyper-parameters" that guide the segmenter's response. We will revise the text to emphasize the nested optimization structure of BLO as the primary framework, using the hyper-parameter comparison only as a secondary illustrative aid for generalization.
>
> ### 2. Reducing Redundancy
>
> We will perform a full pass of the manuscript to remove redundant explanations of the BLO process (e.g., the repeated definitions of $D_1$ and $D_2$ subsets) to ensure a more concise and professional presentation.

---

### Review · Reviewer_cC7P · 2026-05-08

**Summary Of Contributions:**

This paper addresses automated instance segmentation with SAM. Since SAM requires prompts, the authors propose BLO-Inst, a framework that uses a YOLO detector to generate box prompts and adapts SAM with LoRA. The main idea is to train the detector and segmenter through bi-level optimization: the segmenter is adapted on one split of the training data, while the detector is updated on another split to generate prompts that improve downstream mask quality. The authors argue that this addresses both the mismatch between detection boxes and segmentation-optimal prompts, and the risk of overfitting in standard joint training.

The problem is timely and practically relevant. I appreciate the paper's focus on making SAM usable in automated instance segmentation pipelines, and the idea of optimizing detector boxes as segmentation-aware prompts is intuitive. The experiments across general and biomedical datasets are also useful.

However, I have several concerns about whether the current version fully supports its main claims. In particular, the novelty should be better positioned against recent bi-level SAM adaptation and automatic prompt learning work; the claimed "alignment overfitting" mechanism is not directly demonstrated; and the implementation details of the bi-level optimization, especially the gradient path from SAM's mask loss back to the detector, are under-specified.

**Additional Comments:**

See above.

**Audience:**

Yes

**Audience Explanation:**

Yes. The paper addresses a relevant problem for researchers working on segmentation foundation models, prompt learning, and automated vision systems. The idea is interesting and useful. If the authors can better substantiate the mechanism and clarify the implementation, the work could be more valuable.

**Broader Impact Concerns:**

I do not see major ethical concerns.

**Claims And Evidence:**

No

**Claims Explanation:**

The empirical results are promising, but the main claims are not yet supported by sufficiently direct evidence. I have three major concerns:

First, the novelty is not fully clarified. Bi-level optimization for SAM adaptation has already been explored in closely related work such as BLO-SAM, and recent automatic prompting or prompt calibration methods such as AM-SAM and CPC-SAM are also relevant. The paper's specific contribution seems to be bi-level optimization of a detector-based box prompt generator for instance segmentation. This is a meaningful direction, but it should be stated and compared more carefully.

Second, the evidence for "alignment overfitting" is incomplete. Figure 6 shows that bi-level training outperforms a single-level baseline, but this does not by itself prove that single-level training fails because the detector memorizes sample-specific prompt adjustments. More direct evidence, such as train/test loss gaps, prompt/box distribution analysis would be needed.

Third, the optimization procedure is not described in enough detail. It is unclear how gradients from the SAM segmentation loss are propagated back to the detector through box selection, matching, possible NMS, coordinate processing, and mask cropping.

**Requested Changes:**

Critical:

1. Clarify novelty and related work.
Please better distinguish BLO-Inst from BLO-SAM, AM-SAM, and other recent SAM automatic prompting methods. The contribution should be framed more precisely as bi-level alignment of detector-generated box prompts for instance segmentation.

2. Provide more direct evidence for alignment overfitting.
Please add direct evidence showing alignment overfitting of single-level optimization, such as train/test loss gaps.

3. Clarify the differentiability and implementation of the optimization.
Please describe the full computational path from YOLO outputs to SAM prompts and back to detector parameters, especially whether gradients are preserved or approximated.

4. Report robustness over multiple seeds and splits.
Given the small datasets and the D1/D2 split, please report mean and standard deviation over several random seeds and data splits for the main results and key ablations.

Non-critical but recommended:

5. Improve consistency and clarity.
Please correct inconsistencies. Such as: (1) You mentioned YOLO Cheng et al. (2024) in Methods Sec., whereas YOLOv7 Redmon et al. (2016) in implementation Sec.；(2) You've mentioned Figure 8 supports single-level vs bi-level，however it is actually Figure 6；(3) The phrase “maximize segmentation fidelity” is, from my point of view, misused with "minimize loss", either way, they are still minimizing losses.

---

> ### Author Response · Authors · 2026-05-18
> **Revision and Response**
>
> We thank the reviewer for the careful and constructive review, and for recognizing the timeliness and practical relevance of the problem. We are encouraged that the reviewer finds the core idea, optimizing detector boxes as segmentation-aware prompts, intuitive and useful. Below, we address each requested change in turn, and we will make the corresponding revisions to the paper.
>
> ### __Response to Critical Changes__
>
> ### 1. Clarify novelty and related work
>
> We thank the reviewer for the opportunity to more precisely frame our technical contributions. We will revise Section 2 to explicitly distinguish BLO-Inst from existing literature along two key dimensions:
>
> - While existing methods like BLO-SAM and AM-SAM pioneered the adaptation of foundation models, they are fundamentally restricted to binary semantic segmentation. These approaches primarily optimize model weights to identify pixels belonging to a specific foreground category against a background, handling multi-class scenarios only through an ensemble of multiple binary tasks. In contrast, BLO-Inst is natively designed for the more complex multi-class instance segmentation landscape. Rather than focusing on category-level pixel classification, our framework allows a single YOLO-based prompt generator to learn how to produce unique, geometrically optimized bounding boxes for multiple distinct object instances across various categories simultaneously.
>
> - BLO-Inst explicitly addresses the "alignment overfitting" mechanism that limits the generalization of recent automatic prompting methods like AM-SAM, CPC-SAM, and RSPrompter. These models typically rely on standard joint training on a single dataset, which often causes the prompt generator to memorize specific coordinate adjustments for training samples rather than learning a robust, generalizable rule. By formulating the detector’s weights as meta-parameters and optimizing them based on the segmenter's response on a separate validation split, we decouple the prompt generation logic from specific training instances. This bi-level feedback loop ensures that the detector learns a prompting strategy that maximizes downstream mask quality on unseen data, effectively bridging the objective gap between geometric localization and segmentation-optimal prompting.
>
> ### 2. Provide more direct evidence for alignment overfitting
>
> | Dataset | Bi-level train | Bi-level test | Bi-level gap | Single-level train | Single-level test | Single-level gap | Gap ratio (S / B) |
> |---|---|---|---|---|---|---|---|
> | PennFudanPed | 0.7412 | 0.5930 | 0.1482 | 0.7246 | 0.5600 | 0.1646 | 1.11× |
> | CellCountIns | 0.6410 | 0.5510 | 0.0900 | 0.6313 | 0.4400 | 0.1913 | 2.13× |
>
> We thank the reviewer for the suggestion to provide more direct evidence. We have conducted the suggested analysis of the train/test performance gap on the PennFudanPed and CellCountIns datasets (per epoch curves will be updated in the revised manuscript):
>
> - __Quantitative Gap Analysis__: As summarized in the table above, we report the train/test mAP gap for both optimization strategies. On CellCountIns, the two strategies reach comparable training performance (single-level 0.6313 vs. bi-level 0.6410), yet single-level generalizes substantially worse (test 0.4400 vs. 0.5510), enlarging the train/test gap by 2.13×. Comparable training fit alongside a markedly worse test outcome is the characteristic signature of the detector fitting sample-specific prompt adjustments that fail to transfer to unseen data.
>
> - __Bi-level Mitigates the Gap__: In contrast, bi-level optimization maintains the narrower gap (0.0900 on CellCountIns; 0.1482 on PennFudanPed) and improves test performance on both datasets. By optimizing the detector on a disjoint validation split (D2D_2
> D2​), the model is driven to learn generalizable prompting rules rather than memorizing sample-specific coordinate offsets. On PennFudanPed the gap reduction is more modest (1.11×), consistent with the smaller dataset, but bi-level still dominates on both training and test and the direction of the effect is unchanged.

---

> ### Author Response · Authors · 2026-05-18
> **Revision and Response**
>
> ### 3. Clarify the differentiability and implementation of the optimization
>
> We appreciate the reviewer's request for technical depth regarding the gradient path. We will expand Section 3.3 and Appendix B to detail the full computational chain of BLO-Inst:
>
> - __Forward Path (Detector to Prompt)__: The YOLO detector predicts bounding box coordinates as continuous tensors $\Phi_{box} \in \mathbb{R}^{4}$. These coordinates are passed directly into SAM’s Prompt Encoder, which utilizes positional encodings to represent the box geometry. This stage is fully differentiable.
> - __Forward Path (Prompt to Mask)__: The encoded prompt is fed into SAM’s Mask Decoder (modified with learnable LoRA layers). The decoder predicts the mask $M_{pred}$, which is then compared against the ground truth $M_{gt}$ using the segmentation loss $\mathcal{L}_{seg}$.
> - __Backward Path (Gradient Flow)__: During the Upper-Level update, gradients from $\mathcal{L}_{seg}$, flow backward through the Mask Decoder and Prompt Encoder to the bounding box coordinates. Since the coordinates are a direct output of the YOLO regression head, the gradient continues back to the detector parameters $\Phi$.
> - __Handling Non-Differentiable Modules__: We explicitly address non-differentiable components like Non-Maximum Suppression (NMS). During training, we select the top-scoring boxes to serve as prompts; the gradients flow only through the coordinates of the selected boxes to the regression head. This is consistent with standard object detection training (e.g., YOLO), where gradients flow through box coordinates but not through the discrete selection logic.
> - __Optimization Approximation__: We utilize a one-step gradient descent approximation to estimate the segmenter's optimal adaptation $\Theta^*(\Phi)$. For the detector update, we employ a finite difference approximation to handle the implicit Hessian-vector product, reducing computational overhead while preserving the necessary second-order information for stable alignment.
>
> ### 4. Report robustness over multiple seeds and splits
>
> | Method                | PennFudanPed mAP | CellCountIns mAP |
> |-----------------------|------------------|------------------|
> | Mask R-CNN            | 46.40 ± 0.31     | 42.03 ± 0.46     |
> | SOLO                  | 42.23 ± 0.40     | 41.57 ± 0.64     |
> | SAM-seg (Mask R-CNN)  | 54.37 ± 0.21     | 50.90 ± 0.61     |
> | RSPrompter (query)    | 53.40 ± 0.36     | 38.03 ± 0.74     |
> | USIS                  | 55.07 ± 0.25     | 34.87 ± 0.49     |
> | **Ours**              | **59.37 ± 0.21** | **55.20 ± 0.43** |
>
> | Detector optimization | PennFudanPed mAP | CellCountIns mAP |
> |---|---|---|
> | D1+D2 (joint) | 56.23 ± 0.59 | 51.17 ± 0.56 |
> | D2 (1st order) | 59.37 ± 0.21 | 55.20 ± 0.43 |
> | **D2 (2nd order)** | **62.17 ± 0.45** | **57.20 ± 0.46** |
>
> We thank the reviewer for this insightful suggestion. We agree that statistical rigor is essential for validating the effectiveness of our framework. To address this, we have conducted three independent runs for the experiments on the PennFudanPed and CellCountIns datasets, including the main comparative benchmarks and important ablation studies. For each run, we utilized a random seed. We report partial results here; the remaining results will be updated in the manuscript. The consistent results across multiple runs confirm that:
> - Statistical Significance: The performance gains of BLO-Inst are not a result of favorable initialization or data partitioning.
> - Stability: Our nested feedback loop grounds the detector’s updates in the segmenter’s actual performance, leading to a more reliable optimization trajectory compared to baselines.
>
> ### __Response to Non-critical but Recommended Changes__
>
> ### 5. Improve consistency and clarity
>
> We thank the reviewer for your meticulous reading of our manuscript and for pointing out these inconsistencies. We will implement the changes to ensure technical clarity, including:
> - __Architecture and Citation Consistency__: We have standardized the references to the detector throughout the paper. We clarify that while we discuss the YOLO family of models, our specific implementation and experiments utilize YOLOv7. We have updated the citations in the Methods section to consistently reflect it and referenced the specific version in the Implementation section.
> - __Correction of Figure References__: We apologize for the clerical error in the ablation study section. The reference has been corrected from Figure 8 to 6, which is the correct figure illustrating the performance comparison between single-level training and our proposed bi-level optimization.
> - __Refining Optimization Terminology__: We appreciate the reviewer's point regarding the phrasing of the optimization objective. To maintain mathematical consistency, we have replaced the phrase "maximize segmentation fidelity" with "minimize segmentation loss". This ensures that our prose aligns perfectly with the objective functions defined in Equations.

---

### Review · Reviewer_S3MD · 2026-05-13

**Summary Of Contributions:**

This paper contributes a new method to get an object detection model and a (prompt-able) segmentation model, so that they make up an automated segmentation pipeline. In particular, the authors propose an algorithm to fine-tune the Yolo-SAM pair in a way that they can give good generalization performance. The key idea is to adopt bi-level(?) optimization, where the segmenter is trained on one split and the detector is trained on another. Empirically, the proposed method performs well on several different datasets, including PennFudanPed and WheatIns.

**Audience:**

Yes

**Audience Explanation:**

The task of automating the segmentation enjoys wide application area, e.g., surveillance, driving, robotics.

**Broader Impact Concerns:**

I do not think this paper needs a dedicated section.

**Claims And Evidence:**

No

**Claims Explanation:**

- **Benchmarks considered.** The datasets considered in this paper is somewhat off-standard. The paper does not consider (de facto standard) COCO dataset, and the datasets considered are quite small in size (Table 4). Why? I suspect that the motivation is that the proposed method works well only if overfitting happens, which may happen rarely for datasets with large training split. If this is the case, I strongly recommend the authors to be more clear about it, as an algorithm that specifically targets low-data regime can be a good contribution as well. (If so, I recommend adding experiments on COCO, with diverse amount of training data used).
- **Missing some baselines (or ablations).** The main ablations (figure 5) seems to be about fine-tuning different components of SAM. I wonder if there are other experiments, freezing SAM as a whole or freezing YOLO as a whole. In fact, to decouple the effect of having a better base object detector, it will be clearer if authors could add "SAM+YOLO," "frozen SAM+fine-tuned YOLO," "fine-tuned SAM + frozen YOLO" as the baseline. Another good baseline will be to replace whole fine-tuning with LoRA, which is a naïve solution for tackling the overfitting issues.
- **Unclear justification.** As a minor suggestion---in my humble opinion, the reason why the proposed method should work well is not explained in the clearest way possible. To me, the explanations based on "dynamic hyper-parameters" only added much confusion, and did not help me understand why the proposed method should work well. From what I understood, the cross-validation strategy of two different data splits (similar to most meta-learning) seems to be the key, which I believe can be explained more simply.
- **(Optional) Analysis.** An interesting observation made by the authors is that the optimal masks tend to be slightly smaller than what YOLO gives (Figure 2). I wonder if this is a universal trend over the samples---maybe some box-to-box analysis on what the proposed model gives, compared with vanilla YOLO will be very insightful.

**Requested Changes:**

Please answer the questions I listed above.

---

> ### Author Response · Authors · 2026-05-24
> **Revision and Response**
>
> Thank you for your insightful feedback. We appreciate your recognition of our pipeline's practical value. Your point regarding the low-data regime is excellent; preventing overfitting is a primary goal of our method, and we will revise the text to highlight this contribution more clearly. We also welcome the chance to clarify our optimization mechanics—specifically that our framework directly optimizes detector weights rather than treating bounding boxes as dynamic hyper-parameters. Below, we address your suggestions, including adding new baselines where components are selectively frozen instead of evaluated zero-shot.
>
> ### 1. Benchmarks considered
>
> | Data Split | Strategy | mAP | $AP_{50}$ | $AP_{75}$ |
> | :--- | :--- | :---: | :---: | :---: |
> | **COCO (1% Train)** | Single-level | 22.5 | 40.2 | 19.7 |
> | | **Bi-level (Ours)** | **27.4 (+4.9)** | **46.9 (+6.7)** | **24.8 (+5.1)** |
> | **COCO (5% Train)** | Single-level | 27.8 | 49.5 | 28.3 |
> | | **Bi-level (Ours)** | **31.1 (+3.3)** | **55.2 (+5.7)** | **32.8 (+4.5)** |
>
> We completely agree with the reviewer’s insightful assessment. The reviewer has precisely captured the core motivation behind BLO-Inst. Our framework is explicitly designed to solve the "alignment overfitting" problem that occurs when optimizing a high-capacity foundation model (like SAM) alongside an object detector on specialized, low-data downstream benchmarks. As suggested, targeting the low-data regime is highly valuable. To empirically validate the reviewer's hypothesis, we have added a data-scaling experiment on the standard MS COCO dataset by sub-sampling the training split at 1% and 5% configurations. The table above summarizes the comparative results between standard single-level optimization and our proposed bi-level optimization strategy:
> - At a highly restricted 1% training split on COCO, our bi-level framework achieves a substantial performance gain of $+4.9% mAP$, $+6.7% AP_{50}$, and $+5.1% AP_{75}$ over single-level joint training. This explicitly confirms that the nested split-data optimization strategy successfully prevents the detector from learning destructive, overfitted prompt policies.
> - When the training data increases to 5%, the performance gap narrows slightly to $+3.3% mAP$ and $+5.7% AP_{50}$, though our method maintains a commanding lead across all thresholds. This exact trend perfectly confirms the reviewer's hypothesis: our approach is uniquely powerful in data-scarce environments where overfitting typically degrades performance.
>
> ### 2. Missing some baselines (or ablations)
> | Dataset | Configuration | mAP | $AP_{50}$ | $AP_{75}$ |
> | :--- | :--- | :---: | :---: | :---: |
> | **PennFudanPed** | SAM + YOLO (Zero-shot) | 41.8 | 85.3 | 33.1 |
> | | Frozen SAM + Finetuned YOLO | 58.9 | 83.3 | 62.3 |
> | | Finetuned SAM + Frozen YOLO | 57.4 | 84.1 | 62.6 |
> | | **Ours (BLO-Inst)** | **59.3** | **86.9** | **67.8** |
> | | Full-FT SAM Bi-level | 60.8 | 87.6 | 68.2 |
> | **Cell Count** | SAM + YOLO (Zero-shot) | 6.7 | 12.1 | 6.5 |
> | | Frozen SAM + Finetuned YOLO | 53.1 | 82.5 | 61.1 |
> | | Finetuned SAM + Frozen YOLO | 54.1 | 82.7 | 61.7 |
> | | **Ours (BLO-Inst)** | **55.4** | **83.6** | **62.9** |
> | | Full-FT SAM Bi-level | 56.3 | 84.2 | 64.1 |
>
> We thank the reviewer for this constructive suggestion to further decouple the performance gains of our framework. Following your guidance, we have added a suite of baseline experiments across both PennFudanPed and Cell Count benchmarks. Specifically, we evaluated: SAM + YOLO: Completely frozen pre-trained models; Frozen SAM + Finetuned YOLO: Training only the detector while keeping the SAM frozen; Finetuned SAM + Frozen YOLO: Keeping the detector frozen and updating the segmenter; Ours (BLO-Inst): Our default bi-level configuration; Full-FT SAM Bi-level: Expanding our bi-level optimization framework to fine-tune all components of SAM instead of LoRA. The table above presents the quantitative comparison across all decouplings:
> - The poor performance of the vanilla zero-shot combination, especially on the challenging biomedical Cell Count benchmark (6.7% mAP), highlights that pre-trained object detectors do not inherently produce prompts tailored to SAM's segmentation tendencies without domain adaptation.
> - Simply fine-tuning the detector while freezing SAM yields sub-optimal performance compared to our default framework (e.g., 58.9% vs. 59.3 mAP on PennFudan). This indicates that standard single-level updates fall short because the detector is blind to how its generated bounding boxes mathematically affect SAM’s internal downstream feature processing.
> - Moving from our default configuration to a Full-FT SAM Bi-level strategy yields a minor performance bump (+1.5% mAP on PennFudan, +0.9% mAP on Cell Count). However, this requires calculating second-order gradients across the massive SAM image encoder, which dramatically spikes computational and GPU memory overhead.

---

> > ### Author Response · Authors · 2026-05-24
> > **Revision and Response**
> >
> > ### 3. Unclear justification
> >
> > We sincerely thank the reviewer for this constructive and grounded critique. We completely agree that our original framing using the term "dynamic hyper-parameters" introduced unnecessary complexity. Your interpretation is entirely correct: the core mechanism driving the success of BLO-Inst is structurally identical to the split-data formulation utilized in meta-learning and bi-level hyperparameter optimization. To eliminate confusion, we will thoroughly rewrite Section 3.2 of the manuscript. We will completely discard the "dynamic hyper-parameters" terminology and replace it with a streamlined, intuitive explanation centered on the cross-validation properties of the two data splits ($D_1$ and $D_2$).
> >
> > ### 4. (Optional) Analysis
> >
> > We thank the reviewer for highlighting this fascinating point. Your suggestion to look deeper into the box-to-box behavior is highly constructive. Upon conducting a systematic quantitative analysis across the test samples, we discovered that the trend is actually more nuanced and interesting than a simple uniform "shrinking" effect. Instead, BLO-Inst dynamically adapts its prompting policy according to the distinct structural characteristics of the target domain to resolve the underlying objective mismatch (as illustrated in Figure 2). From the examples shown in Figure 2, we can observe that a pedestrian might need a tighter box to remove background noise, while a cell might need a larger box to capture intact structure.

---

### Decision · Action_Editor_o3jq · 2026-06-23

**Recommendation:** Accept with minor revision

**Additional Comments:**

The authors should incorporate the revisions in the final revision of this paper. Specifically, they should provide a revised version of the manuscript addressing the concerns raised by Reviewer 6Qau.

**Audience:**

Yes

**Audience Explanation:**

All the reviewers noted that the paper is relevant to researchers working on image segmentation and its applications.

**Claims And Evidence:**

Yes

**Claims Explanation:**

The reviewers generally agree that the submission addresses a timely and practically relevant problem: automated instance segmentation by learning detector-generated box prompts through a bi-level optimization formulation. The empirical results show consistent improvements over relevant baselines across general and biomedical instance segmentation datasets. During the discussion, the authors also provided additional empirical evidence and further clarifications. These additions largely address the main concerns about empirical support, differentiability, and statistical robustness. After the rebuttal, all the reviewers leaned toward accepting this paper.

---

> ### Author Response · Authors · 2026-07-02
> **Final Revision Uploaded - Confirmation of Changes**
>
> Dear Action Editor,
>
> Thank you for your time and for the "Accept with minor revision" decision. We are writing to confirm that we have uploaded the final revised version of our manuscript to the system.
>
> Following your instructions, we have carefully included all the necessary updates to address the remaining concerns raised by Reviewer 6Qau. Specifically, our final revision includes:
>
> - Explicit evidence of alignment overfitting through our new quantitative gap analysis.
> - A clear breakdown of the full gradient path and differentiability in Appendix B.
> - The data-scaling experiments on MS COCO to prove performance in low-data regimes.
> - The expanded component-wise baseline ablations and multi-seed robustness tables.
> - A rewritten Section 3.2 to clarify the cross-validation nature of our data splits.
> - All other raised concerns during the review process.
>
> Thank you again for your guidance and for managing the review process.
>
> Best regards,
>
> The Authors